# Puzzle Everything: Benchmarking Knowledge Reasoning through Generalized Puzzle Solving

## Abstract

The validity of existing benchmarks is challenged by data leakage, which risks reducing current evaluation from genuine problem-solving to mere recitation. This concern highlights a fundamental need for robust methods to assess reasoning based on a model's internalized knowledge. To address this issue, we introduce the Puzzle Everything Benchmark (PEB), which fundamentally expands the puzzle-solving application for reasoning evaluation. By treating any learned concept, term, or entity as a solvable puzzle, our approach enables a direct assessment of a model's ability to reason over any specialized knowledge. Since it is well-established that Wikipedia serves as a cornerstone of the training corpora for most models, PEB facilitates this with a Wikipedia collection of 960 puzzles of doctoral-level difficulty drawn from eight distinct disciplines. Our evaluation on PEB reveals a counter-intuitive phenomenon: most open-source reasoning models significantly underperform their standard, no-thinking counterparts. This finding suggests that under the interactive evaluation paradigm of PEB, these models exhibit strategic failures when attempting to apply procedural reasoning proficiency to specialized knowledge domains. Thus, the construction of the PEB introduces a scalable method for rapidly generating diagnostic benchmarks in any specialized domain, enabling a robust evaluation of reasoning that is insensitive to prior data exposure.

## 1 Introduction

The rapid evolution of Large Language Models (LLMs) has necessitated the development of robust and reliable evaluation paradigms. However, the validity of the current benchmark is under threat from data leakage and contamination (Li et al., 2023; Jacovi et al., 2023; Roberts et al., 2023), where test items overlap with the undisclosed corpora used for pre-training. This issue poses an increasing threat to the validity of LLM-based research (Xu et al., 2024; Zhou et al., 2025). It effectively corrupts the evaluation process, transforming what should be a test of genuine ability into a mere assessment of recitation (Singh et al., 2024). These persistent issues necessitate a shift towards evaluation paradigms designed to be inherently resistant to data contamination.

A prominent response has been the shift towards dynamic benchmarks (Kiela et al., 2021; Shirali et al., 2022). Approaches like LatestEval (Li et al., 2024) and LiveBench (White et al., 2024) use continuously updated data, but this temporal strategy is a patch-like solution, requiring a constant, contrived pipeline to stay ahead of model training cycles. Other methods like Chatbot Arena (Chiang et al., 2024) produce unstable judgments, as these crowdsourced comparisons are influenced by subjective factors like response length and style rather than objective capabilities. Furthermore, current agent benchmarks (Liu et al., 2023; Zhou et al., 2023; Chan et al., 2024) predominantly focus on domains with clear verification rules, such as coding or web search, leaving the more difficult task of evaluating reasoning on general knowledge largely underexplored. These persistent challenges highlight the difficulty in constructing reliable and scalable dynamic benchmarks.

To address this challenge, puzzle-based evaluations (Giadikiaroglou et al., 2024) offer a promising direction. Puzzles naturally support process-level evaluation, as they require models to explore and reason rather than simply retrieve direct answers. Prior work has often focused on specific

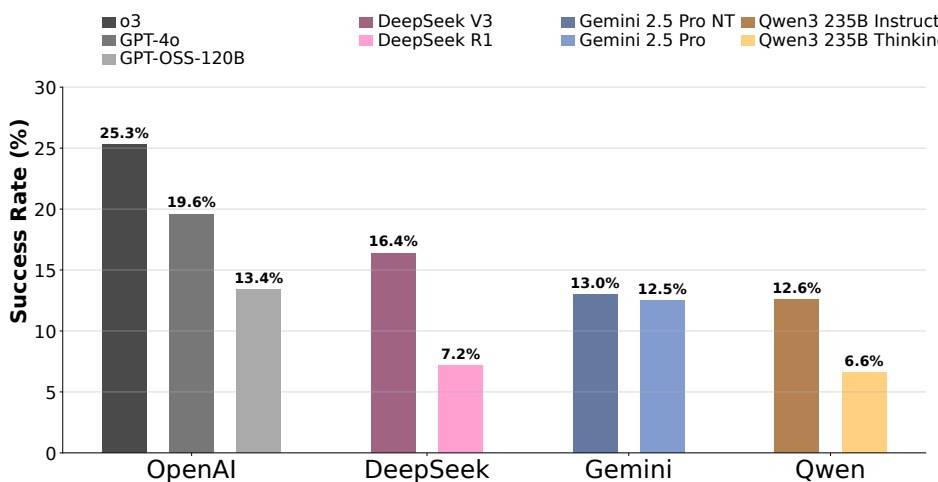

Figure 1: Comparison of success rates between "thinking" and "no-thinking" model variants on the PEB. A widespread and counter-intuitive performance degradation is observed in the open-source reasoning-enhanced models. This figure highlights the direct comparison for these paired models, while comprehensive results for the entire model suite, including others hybrid models like GPT-5 are detailed in Table 1.

puzzle types (Chollet, 2019; Lin et al., 2021; Zhang et al., 2025), such as riddles, Sudoku and other games. Recent benchmarks have advanced this approach. For instance, SPLAT (Chen et al., 2024) frames tasks as clue-seeking within pre-scripted narratives, and MTP (Badola et al., 2025) assesses models through carefully designed interactive environments. However, since such tasks are difficult to generalize and are dominated by external rules, they primarily measure a models capacity to learn new game mechanics rather than its underlying reasoning ability. Our work extends the use of puzzle-solving by integrating it with domain ontological knowledge, aiming instead to bridge this gap and evaluate whether models can demonstrate a generalized, reasoning-driven understanding of established concepts.

In this work, we introduce the Puzzle Everything Benchmark (PEB), which fundamentally expands the scope of puzzle-based reasoning evaluation. Specifically, PEB is designed to assess ontology-aware knowledge reasoningthe capacity to navigate hierarchical conceptual structures through strategic inquiry. This mechanism finds a concrete parallel in high-stakes real-world domains, such as medical diagnosis. Much like a physician systematically narrows down possibilities through multi-turn questioning to identify a disease, PEB requires the model to execute an Ontological Traversal process. To succeed, the model must infer the domain's implicit taxonomic structure and probe entity attributes to formulate queries that efficiently partition the search space and pinpoint the target concept. Leveraging the fact that Wikipedia serves as a cornerstone of pre-training corpora for foundational models, PEB constructs 960 puzzles centered on entities of doctoral-level difficulty. These span eight major academic domains, including STEM, the humanities, and health sciences. This design ensures PEB transcends superficial gameplay mechanics, serving instead as a comprehensive instrument for evaluating the structural organization and reasoning accessibility of expert knowledge within LLMs.

Evaluation of a wide range of leading LLMs on PEB reveals the benchmark's significant challenge where even the top-performing model GPT-5 achieved a success rate of only 33.4% (Table 1). The results uncover a counter-intuitive phenomenon: most open-source reasoning-enhanced models significantly underperform their no-thinking counterparts, a trend exemplified in Figure 1 by models including GPT-OSS-120B, DeepSeek R1, and Qwen3-235B-A22B-Thinking. Detailed analysis of response patterns also highlights distinct failure modes in these models. For instance, Gemini's relative weakness stems from a tendency to frequently offer a conclusive guess which points to potential inherent flaws. These phenomena suggest that under the constraints of interactive evaluation, these models struggle to effectively translate procedural reasoning proficiency into strategic application within specialized knowledge domains.

Beyond these empirical findings, the construction of the PEB introduces a scalable method to rapidly generate diagnostic benchmarks in any specialized domain, enabling a robust evaluation of reasoning that is insensitive to prior data exposure. We further showcase its application by building an IC Back-End Puzzle Benchmark, featuring 480 puzzles from four EDA command tools in the IC back-end design (Appendix B). The core contribution can be summarized as follows:

- We introduce the Puzzle Everything Benchmark (PEB), a challenging suite of 960 doctoral-level puzzles from Wikipedia that expands the application of puzzle-solving to general knowledge reasoning.

- We present a scalable construction methodology derived from the PEB pipeline, for rapidly generating domain-specific benchmarks that are inherently robust against data leakage.

- We identify a critical proficiency gap in leading open-source reasoning models, observing that under the interactive evaluation paradigm of PEB, these models exhibit distinct strategic failures when applying procedural reasoning to knowledge-driven tasks.

## 2 RELATED WORK

### 2.1 BENCHMARK EVALUATION AND THE CHALLENGE OF DATA LEAKAGE

The evaluation of large language models has been largely driven by a suite of comprehensive benchmarks designed to assess their knowledge and reasoning abilities across a multitude of domains (Cobbe et al., 2021; Liu et al., 2024b; Rein et al., 2024; Clark et al., 2018; Hendrycks et al., 2021). Prominent examples such as MMLU (Hendrycks et al., 2020), BIG-Bench (Srivastava et al., 2023) and HellaSwag (Zellers et al., 2019) have become standard tools for measuring model performance (Contributors, 2023; Duan et al., 2024). A common characteristic of these benchmarks is their reliance on static, single-turn formats, typically involving multiple-choice questions or direct answer generation (Wang et al., 2024b; Suzgun et al., 2022; Yue et al., 2024; Lu et al., 2022). However, static benchmarks are vulnerable to data leakage, since portions of their test data may inadvertently be included in the massive datasets used for model pre-training (Balloccu et al., 2024; Xu et al., 2024; Ni et al., 2025; Li et al., 2023). One way to mitigate cheating is through dynamically updated benchmarks, such as LatestEval (Li et al., 2024) and LiveBench (White et al., 2024), which continuously incorporate the latest real-world data. While this approach alleviates the issue to some extent, it still fails to enable fine-grained analysis and precise evaluation of the internal knowledge a model has acquired. To address this, we propose a dynamic, multi-turn framework designed to preclude rote memorization and compel the model to externalize its deductive reasoning.

### 2.2 PUZZLES AND GAMES AS EVALUATION PARADIGMS

Puzzles, broadly defined as structured problems that require reasoning to reach a solution, have long served as effective probes of intelligence (Guo et al., 2024; Estermann et al., 2024; Shah et al., 2024). They constrain the search space while still demanding creativity, logical inference, and the integration of disparate knowledge. Accordingly, puzzles and games have been widely adopted as formal evaluation paradigms, from classic board games like Chess and Go (Silver et al., 2017) to text-based adventure games (Wang et al., 2024a; Gallotta et al., 2024; Zhang & Long, 2025) and interactive formats such as 20 Questions (Vinyals et al., 2019; Liang et al., 2023; Hu et al., 2018). A common feature of these settings is sustained, multi-turn interaction within a structured environment, which makes them well-suited for assessing strategic planning, logical deduction, and efficient use of information (Chen et al., 2024; Li et al., 2025; Chen et al., 2025). Motivated by this tradition, we frame evaluation as a multi-turn, zero-prior game that requires a model to externalize its reasoning trajectory rather than merely produce final answers. Unlike traditional game-based evaluations that focus on optimizing win-loss strategies against an adversary (Nowé et al., 2012; Silver et al., 2017), the Puzzle Everything framework repurposes the puzzle format to evaluate how a model retrieves, integrates, and reasons over its internalized knowledge to identify a target concept it is expected to know, thereby prioritizing the interpretability and logical coherence of the reasoning trajectory over simple binary success metrics.

---

**Round 0, Puzzle: Reticular Dysgenesis(is a rare, inherited autosomal recessive disease...)**

You are a human disease expert. I will have a preset human disease name/term/concept in mind, and you need to continuously ask questions ...Your questions must be in the form of a single, four-option multiple-choice question at a time.
I have now preset a name/term/concept. Please begin with your first question.

---

**Round 1 ~ n, n <= MAX_ROUNDS**

[Round 1]
I'll help you identify the disease. Let me start with a broad categorization question.
Question 1: Which body system is primarily affected by this disease?
A) Cardiovascular system B) Respiratory system C) Nervous system D) Endocrine/Metabolic system
**<pure_parse>** A is incorrect, B is incorrect, C is incorrect, D is correct **</pure_parse>**

[Round 2]
So we're dealing with an endocrine/metabolic disorder. Let me narrow this down further.
Question 2: Which one best describes the primary metabolic abnormality in this disease?
A) Abnormal glucose/sugar metabolism B) Abnormal thyroid hormone levels C) Abnormal lipid/fat metabolism D) Abnormal mineral metabolism
**<pure_parse>** A is incorrect, B is incorrect, C is incorrect, D is incorrect **</pure_parse>**
...

[Round 8]
I need to reconsider my approach. You confirmed it's... Let me think about other metabolic causes of SCID.
Question 8: Is this disease:
A) Adenosine kinase deficiency B) Nucleoside phosphorylase deficiency affecting pyrimidines C) A disorder of folate metabolism affecting nucleotide synthesis D) Reticular dysgenesis (a form of SCID with metabolic component)
**<pure_parse>** A is incorrect, B is incorrect, C is incorrect, D is correct **</pure_parse>**
**<pure_answer>** Reticular dysgenesis **</pure_answer>**

---

Figure 2: A Human Disease sample from the PEB multi-turn evaluation where the agent successfully identifies the target concept, 'Reticular dysgenesis'. The agent's coarse-to-fine questioning strategy externalizes its deductive reasoning path, making the entire trajectory the object of evaluation.

## 3 METHOD

PUZZLE EVERYTHING BENCHMARK is founded on a fundamental generalization of the puzzle definition, which enables the rapid construction of custom reasoning benchmarks. Section 3.1 details this generalized puzzle setting, along with our evaluation methodology and metrics. Section 3.2 then describes the construction details in the PEB.

### 3.1 MULTI-TURN STRATEGIC ASSESSMENT

Our evaluation method is a multi-turn strategic assessment protocol. Its extensibility is derived from the highly generalizable definition of puzzle, as detailed below.

**Generalized Puzzle Concept** Central to our framework is the generalization of the puzzle concept. We move beyond the traditional notion of a puzzle as a contrived game and instead treat any preset concept, term, or entity as a solvable puzzle. In practice, this means any Wikipedia title or technical command can serve as the $P_{target}$ for an evaluation task.

Figure 3: The multi-stage, coarse-to-fine data filtering pipeline used to construct the Puzzle Everything Benchmark from the English Wikipedia dump.

**Multi-turn Strategic Evaluation** The PEB evaluation protocol places the model in a dynamic, non-leaking interactive environment, requiring it to act as a strategic agent to identify the target concept $P_{\text{target}}$ through multi-turn querying.

**Problem Formulation.** We model the domain knowledge as a graph $G = (V, E)$, where $V$ is the set of concepts (entities) and $E$ represents taxonomic relationships. The agent's goal is to identify $P_{\text{target}}$ within a limited budget $T$ by generating a sequence of queries $Q = \{q_1, q_2, \ldots, q_T\}$. At time step $t$, the search space $S_t \subseteq V$ represents the subset of entities consistent with the interaction history. Initially, the search space $S_0 = V$.

**Interaction Mechanism.** Specifically, the agent proposes a query in each round based on the entire history of previous interactions. The query can be a four-option multiple-choice question or a direct guess of $P_{\text{target}}$. The environment agent consults the reference document ($I_{\text{ref}}$) for $P_{\text{target}}$ to provide only correctness feedback on the query (e.g., 'A is correct, B is incorrect...'). Success is achieved when the strategic agent names $P_{\text{target}}$ or one of its aliases from $I_{\text{ref}}$; failure occurs if the MAX_ROUNDS ($T$) limit is reached.

**Search Efficiency and Strategy.** The multi-turn protocol compels the agent to synthesize all interactions to build and refine a dynamic hypothesis. Successful convergence depends on the agent selecting attributes that significantly partition the search space, ensuring that the next candidate set $|S_{t+1}|$ is much smaller than the current set $|S_t|$ ($|S_{t+1}| \ll |S_t|$).

### 3.2 THE PUZZLE EVERYTHING BENCHMARK

To assess the specialized knowledge that models internalize from core training corpora like Wikipedia, we constructed the **Puzzle Everything Benchmark**, featuring 960 puzzles derived from Wikipedia across eight distinct domains.

**Data Construction** The benchmark's puzzles were curated through a rigorous, multi-stage data filtering pipeline, which is illustrated in Figure 3. This process was designed as a coarse-to-fine funnel aimed at distilling highly specialized concepts of various disciplines from a complete Wikipedia dump.

Initially, a corpus of over six million article titles was screened to form a candidate pool requiring specialized knowledge. This pool was subsequently refined to yield a collection of titles at a doctoral-level difficulty or greater. Following this coarse-to-fine selection, the advanced titles were categorized into their respective academic disciplines. The final selection stage employed an intra-disciplinary, percentage-based difficulty screening to identify 120 $P_{\text{target}}$ from each respective domain. This rigorous pipeline resulted in a strict 0.14% retention rate. To further valid the "doctoral-level" difficulty, we conducted both a topological analysis of the knowledge graph and a human expert audit, which confirmed that the selected entities represent highly specialized "edge nodes" rather than common concepts. Comprehensive details on the construction pipeline and difficulty validation are provided in Appendix G. For each puzzle, the cleaned text from the corresponding article serves as the golden reference document, $I_{\text{ref}}$.

**Benchmark Composition** The Puzzle Everything Benchmark is composed of 960 puzzles, which are systematically categorized into eight distinct domains: Biology, Human Disease, Chemistry, Physics, Mathematics, Computer Science, Economics, and Philosophy. These domains were chosen to ensure comprehensive coverage of major academic fields, including the humanities and social sciences, the natural and engineering sciences, and health.

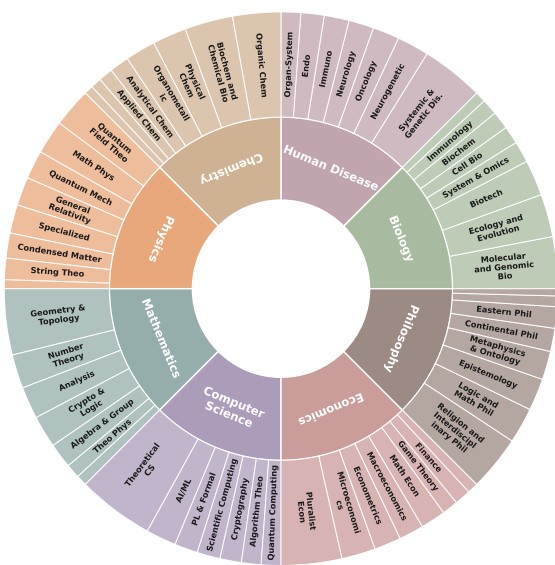

Figure 4: Hierarchical distribution of puzzle targets across domains and sub-disciplines. Eight distinct color families represent the primary domains, while individual segments denote specific subtopics. The area of each segment is proportional to the number of questions in that category. Exact distributions are provided in Appendix H.

Each domain comprises 120 puzzles distributed across 7 to 8 primary sub-disciplines (Figure 4) to ensure diverse topical coverage. For example, the Physics domain sources items from eight distinct subfields, with key contributions from Quantum Field Theory and Mathematical Physics. We also provide additional granularity within each subfield. For instance, the "Specialized Topics" sub-discipline covers Plasma Physics, Atomic, Molecular, and Optical (AMO) Physics, and Particle and Nuclear Physics. This multi-topic, fine-grained setup is a fundamental aspect of our benchmark across all domains, supporting a comprehensive evaluation of expert knowledge. Full details for all domains appear in the Appendix H.

## 4 EXPERIMENTS AND ANALYSES

### 4.1 SETUP

This section outlines the key parameter configurations and the set of models evaluated in our experiments.

**Maximum Rounds.** We set MAX_ROUNDS to 15. This decision is justified by empirical data (Table 2), which shows that a successful session concludes in 9.78 turns on average (9.15 for the top-performing GPT-5). A limit of 15 thus provides sufficient opportunity for models to converge without being cost-prohibitive. Simultaneously, this cap manages the escalating token costs associated with including the full conversation history in each turn. While increasing the limit to 20 would have nearly doubled the evaluation cost.

**Temperature.** All models were configured with a temperature of 0.2 to prioritize evaluation stability and reproducibility. The only exception was GPT-5, which no longer supports explicit temperature adjustments and was evaluated using its default "medium" reasoning effort mode.

**Environment Agent.** We designated DeepSeek V3 as the environment agent responsible for providing correctness feedback. This model was chosen for its strong instruction-following capabilities and consistent outputs at a low temperature. To ensure our main findings are not sensitive to this specific choice, we conducted a detailed ablation study (Appendix C). This includes both a subset analysis and a comprehensive cross-validation using Claude Sonnet 4 on the full 960-puzzle

Table 1: Success Rate (%) for various models across eight domains. The best performance in each domain is highlighted in **bold**. The bottom row, 'Domain Avg.', summarizes the average performance for each domain.

| Model | Bio | Dis | Chem | Phys | Math | CS | Econ | Phil | Avg. |
|---|---|---|---|---|---|---|---|---|---|
| GPT-5 | **34.2** | 37.5 | **33.3** | **34.2** | **34.2** | **25.0** | 39.2 | **30.0** | **33.4** |
| Claude Opus 4 | 27.5 | **40.8** | 29.2 | 26.7 | 18.3 | 19.2 | 31.7 | 27.5 | 27.6 |
| Claude Sonnet 4 | 29.2 | 35.0 | 23.3 | 23.3 | 17.5 | 18.3 | **42.5** | 21.7 | 26.4 |
| o3 | 28.3 | 33.3 | 21.7 | 29.2 | 20.0 | 10.0 | 35.0 | 25.0 | 25.3 |
| GPT-4o | 26.7 | 24.2 | 20.0 | 17.5 | 11.7 | 10.8 | 25.0 | 20.8 | 19.6 |
| Grok-4 | 14.2 | 30.8 | 16.7 | 13.3 | 10.0 | 13.3 | 20.8 | 21.7 | 17.6 |
| DeepSeek V3 | 21.7 | 22.5 | 13.3 | 15.0 | 11.7 | 11.7 | 18.3 | 16.7 | 16.4 |
| GPT-OSS-120B | 19.2 | 21.7 | 12.5 | 10.8 | 5.8 | 8.3 | 20.8 | 8.3 | 13.4 |
| Gemini 2.5 Pro No-Thinking | 9.2 | 28.3 | 12.5 | 6.7 | 5.8 | 7.5 | 16.7 | 17.5 | 13.0 |
| Qwen3-235B-A22B-Instruct | 5.8 | 27.5 | 5.0 | 11.7 | 5.8 | 10.0 | 15.0 | 20.0 | 12.6 |
| Gemini 2.5 Pro | 7.5 | 26.7 | 11.7 | 13.3 | 1.7 | 8.3 | 17.5 | 13.3 | 12.5 |
| DeepSeek R1 | 5.8 | 21.7 | 5.0 | 1.7 | 5.0 | 3.3 | 10.0 | 5.0 | 7.2 |
| Qwen3-235B-A22B-Thinking | 7.5 | 11.7 | 5.8 | 3.3 | 1.7 | 5.0 | 9.2 | 8.3 | 6.6 |
| **Domain Avg.** | 18.2 | **27.8** | 16.2 | 15.9 | **11.5** | 11.6 | 23.2 | 18.1 | **17.8** |

benchmark, confirming that the relative performance hierarchy of models remains stable ($\rho = 0.96$) despite absolute scoring shifts.

**Evaluated Models.** Our evaluation on the PEB covers a comprehensive suite of 13 leading closed-source and open-source LLMs to assess their capabilities. Based on the PEB, we conduct a comprehensive assessment of their capabilities. We use the default version of their API for all models (unless otherwise specified). We list all the evaluated models below:

- **Closed-Source models**: GPT-5, o3 and GPT-4o (Hurst et al., 2024), Claude Opus 4 and Claude Sonnet 4, Gemini 2.5 Pro and Gemini 2.5 Pro No-Thinking (Comanici et al., 2025), Grok-4.
- **Open-Source models**: Deepseek V3 (Liu et al., 2024a) and Deepseek R1 (Guo et al., 2025), GPT-OSS-120B (Agarwal et al., 2025), Qwen-235B-A22B-Instruct and Qwen-235B-A22B-Thinking (Yang et al., 2025).

## 4.2 MAIN RESULTS

The main evaluation results are presented in Table 1, covering detailed success rate assessments across 8 domains: Biology (Bio), Human Disease (Dis), Chemistry (Chem), Physics (Phys), Mathematics (Math), Computer Science (CS), Economics (Econ), and Philosophy (Phil).

**Analysis of Main Results.** The evaluation results demonstrate pronounced disparities across 8 domains (Table 1). In the strongest-performing domains, Human Disease (27.8%) and Economics (23.2%) achieved the highest average success rates. The high accuracy in Human Disease can be attributed to its comprehensive and well-structured classification system, while the strong performance in Economics may stem from the relatively lower depth and complexity of its specialized knowledge. In the most challenging domains, Mathematics (11.5%) and Computer Science (11.6%) recorded the lowest success rates, where even GPT-5, the strongest model, attained only 34.2% and 25.0% respectively. Intermediate performance was observed in Biology, Chemistry, Physics, and Philosophy, where success rates fell between these two extremes.

From a model-centric perspective, the results reveal a clear performance hierarchy. While GPT-5 leads the pack, its predecessor GPT-4o (19.6%) demonstrates remarkable longevity, effectively separating the top-performing models and the rest. When compared to other closed-source offerings, the relatively modest results from the Gemini series point to specific weaknesses for this reasoning task. Furthermore, the struggles of DeepSeek R1 (7.2%) and Qwen3-235B-A22B-Thinking (6.6%) underscore the current limitations and vulnerabilities within the open-source reasoning model landscape.

Table 2: Average Performance Metrics for Various Models on all domains. 'PLength' is 'Probing Length'. 'IGuessR', 'IOptionR', and 'IStreakR' are 'Incorrect Guess Rounds', 'Incorrect Option Rounds', and 'Incorrect Streak Rounds' respectively. 'TotalR' and 'SuccessR' are 'Success Rounds' and 'Total Rounds'.

| Model | PLength | IGuessR | IOptionR | IStreakR | SuccessR | TotalR |
|---|---|---|---|---|---|---|
| GPT-5 | 299.88 | 0.32 | 6.16 | 3.61 | 9.15 | 13.02 |
| Claude Opus 4 | 494.29 | 0.48 | 6.41 | 4.11 | 10.23 | 13.60 |
| Claude Sonnet 4 | 564.54 | 0.44 | 5.13 | 3.14 | 9.81 | 13.54 |
| o3 | 356.97 | 0.52 | 6.32 | 3.59 | 9.66 | 13.67 |
| GPT-4o | 472.98 | 0.81 | 6.57 | 4.34 | 10.68 | 14.06 |
| Grok-4 | 1930.48 | 1.15 | 4.87 | 3.08 | 9.72 | 14.04 |
| DeepSeek V3 | 551.12 | 2.29 | 5.31 | 2.68 | 9.96 | 14.07 |
| GPT-OSS-120B | 409.17 | 1.21 | 6.13 | 3.64 | 9.53 | 14.23 |
| Gemini 2.5 Pro No-Thinking | 908.49 | 2.12 | 5.51 | 3.82 | 10.48 | 14.37 |
| Qwen3 235B Instruct | 1396.29 | 1.43 | 7.15 | 4.75 | 9.20 | 14.21 |
| Gemini 2.5 Pro | 906.41 | 2.20 | 5.28 | 3.70 | 10.31 | 14.37 |
| DeepSeek R1 | 574.22 | 5.13 | 3.63 | 2.22 | 10.20 | 14.66 |
| Qwen3 235B Thinking | 1494.57 | 2.98 | 4.15 | 2.91 | 8.21 | 14.48 |
| **Avg.** | 796.88 | 1.62 | 5.59 | 3.51 | 9.78 | 14.02 |

## 4.3 Detailed Analyses

This section provides a detailed analysis of model performance based on Table 2, focusing on the behavioral patterns observed during the reasoning process.

**Probing Length.** The Probing Length (PLength) measures the average character length of the agent's output per turn. This output typically includes not only the multiple-choice question but also the model's explicit reasoning based on prior rounds. Our analysis reveals that increased verbosity does not correlate with better performance on this task. Indeed, efficiency in communication appears to be a hallmark of the top models like GPT-5 (299.88). This trend generally holds for the other top-performing models with relatively concise probing lengths. Furthermore, we observe that PLength is often strongly correlated with a model's developer family, suggesting that different series of models possess inherent and consistent styles for expressing their reasoning process in this interactive setting. For instance, both Gemini models exhibit a similarly high verbosity (PLength $\approx$ 907), as do the DeepSeek models (PLength $\approx$ 560) and the OpenAI models ($299 <$ PLength $< 500$). This suggests that different series of models possess inherent and consistent styles for expressing their reasoning process in this interactive setting.

**Incorrect Guess Rounds.** The Incorrect Guess Rounds (IGuessR) metric quantifies the average number of times an agent makes a conclusive but incorrect guess of the target concept per session. We interpret a high IGuessR as an indicator of model overconfidence, where the agent prematurely commits to a wrong answer despite having insufficient information. Our results show a strong negative correlation between IGuessR and Success Rate, suggesting that more cautious, information-seeking strategies are superior. For instance, the top-performing model GPT-5 has the lowest IGuessR (0.32), while the struggling DeepSeek R1 has the highest (5.13). From a strategic standpoint, such actions represent a low-utility approach by invalidating only a single hypothesis. In a context like Human Disease, a propensity for making frequent, incorrect assertions is a harmful failure mode that could lead to detrimental outcomes, underscoring the importance of penalizing such overconfident behavior in model evaluation.

**Incorrect Probing and Error Correction.** We analyze two metrics related to probing errors: Incorrect Option Rounds (IOptionR) and Incorrect Streak Rounds (IStreakR). IOptionR measures the average number of rounds where all four proposed options are incorrect, indicating a misjudgment in the probing direction. IStreakR measures the longest consecutive sequence of such rounds, reflecting the model's ability to recover from a flawed reasoning path. However, these metrics are not simple indicators of poor performance. An 'all-incorrect' round can be highly informative for a

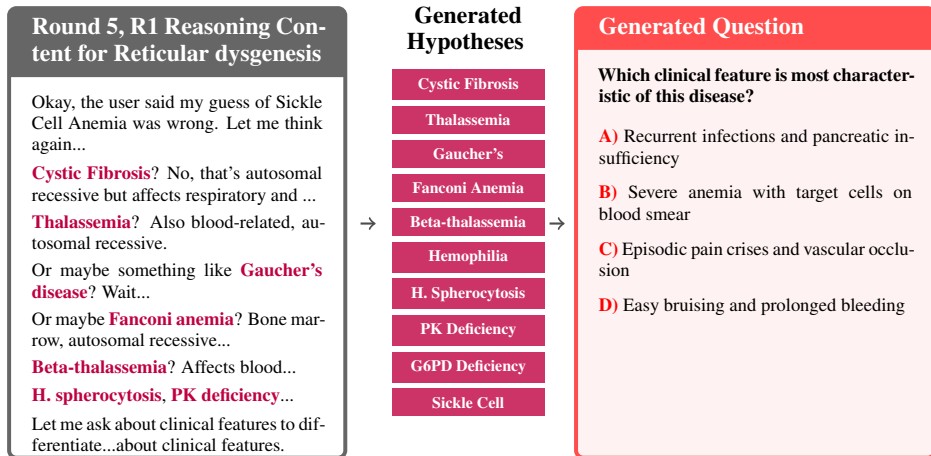

Figure 5: Inefficient "concrete-to-abstract" reasoning in DeepSeek R1 for puzzle Reticular dysgenesis. **Left:** The internal reasoning shows premature generation of numerous specific disease hypotheses after an incorrect guess. **Center:** The extracted specific hypotheses demonstrate scattered, unfocused thinking across multiple disease categories. **Right:** The resulting question generation reflects this scattered approach, potentially missing the optimal diagnostic pathway for the $P_{target}$ 'Reticular dysgenesis'. This inefficient reasoning pattern contrasts with more systematic approaches that would first establish broader diagnostic categories before narrowing to specific conditions.

capable agent, as it allows for significant pruning of the search space and a major strategy correction. This can paradoxically help a strong model quickly locate an obscure concept. Thus, these metrics offer insights into a model's strategic behavior. The case of DeepSeek R1 is particularly illustrative: despite having the lowest IOptionR (3.63) and IStreakR (2.22), it achieves one of the lowest success rates. This suggests that its probing strategy is ineffective at leveraging information and asking questions to find the $P_{target}$, a strategic failure that helps explain its high rate of premature guesses, as reflected in its top-ranking IGuessR.

**Overall Efficiency.** We evaluate efficiency using two complementary metrics: Average Success Rounds (SuccessR) and Average Total Rounds (TotalR). SuccessR measures the average number of rounds in successful sessions only, reflecting speed on solvable problems. TotalR provides a more holistic score by including failed sessions, which are counted as `MAX_ROUNDS` (15), thus blending success rate with efficiency. Relying solely on SuccessR can be misleading for models with low success rates, as it only reflects performance on easier tasks they could solve. TotalR, therefore, offers a more comprehensive view. For example, GPT-5's top performance gives it the best TotalR (13.02) and leading SuccessR (9.15), confirming its position as the most effective model overall. This combined metric also reveals nuances in reasoning efficiency. For instance, despite Claude Opus 4 having a higher Success Rate than Claude Sonnet 4, Sonnet's lower TotalR (13.54 vs. 13.60) and SuccessR (9.81 vs 10.23) suggest it has a relative advantage in the efficiency of its reasoning process.

### 4.4 ERROR ANALYSIS

**Open-Source Reasoning Models.** To understand the relatively weak performance of open-source reasoning models, we analyzed the publicly available chain-of-thought data for GPT-OSS-120B, DeepSeek R1, and Qwen3-235B-A22B-Thinking. Our analysis reveals a recurring flawed strategy: these models tend to generate numerous specific guesses for the final $P_{target}$ within their internal reasoning process, even in the early rounds (Figure 5). They then formulate their multiple-choice questions retroactively based on these premature guesses. This "concrete-to-abstract" approach is diametrically opposed to the "abstract-to-concrete" deductive path required by our task. When the correct $P_{target}$ is not among these initial hypotheses, the model becomes mired in an inefficient reasoning loop filled with irrelevant terms. This flawed process directly contributes to a higher rate of incorrect final guesses, as evidenced by the comparatively high IGuessR values for these models.

**Gemini Series Models.** The Gemini 2.5 Pro models exhibit a distinct behavioral pattern that helps explain their relatively weak performance. A primary characteristic is their high Probing Length (PLength $\approx$ 907), far exceeding other top closed-source models. We observe that this verbosity stems from a tendency to extensively summarize and draw inferences from all prior rounds before posing the question for the current turn. While seemingly thorough, this approach appears to lock the model into a predetermined reasoning path. The continuous summarization of its own previous outputs can lead to a neglect of objective feedback, causing the model to become stubbornly fixed on an incorrect line of inquiry. This strategic rigidity fosters a misplaced confidence, which manifests as a notably high IGuessR (2.1 ~ 2.2), far exceeding that of other leading closed-source models.

## 5 DISCUSSION

Our evaluation on PEB uncovers a counter-intuitive phenomenon: reasoning-enhanced models often struggle with knowledge-driven strategic tasks due to a pervasive "Concrete-to-Abstract" fallacy. Instead of the systematic information gathering observed in standard models, these agents tend toward premature commitment, generating specific, low-probability hypotheses (hallucinated leaf nodes) in an inefficient linear search ($O(N)$). This failure mode contrasts sharply with the optimal "coarse-to-fine" strategy, which we characterize as *Ontological Traversal*—the ability to infer and navigate the implicit taxonomic structure of a domain. We provide a theoretical model of these search dynamics in Appendix D.

This distinction positions PEB in a unique niche orthogonal to existing benchmarks. While static evaluations (e.g., MMLU) assess factual retrieval and procedural tasks (e.g., GPQA) evaluate algorithmic logic, PEB measures *Strategic Knowledge Reasoning*—the dynamic application of expert knowledge under uncertainty. By mirroring high-stakes real-world scenarios like medical diagnosis, where experts must narrow hypothesis spaces through discriminatory questioning, PEB provides a diagnostic view into how models structure and access their internalized information, particularly within the "tail" of specialized knowledge.

## 6 CONCLUSION

Our primary contribution is a new methodology of Generalized Puzzle Solving, instantiated in the Puzzle Everything Benchmark (PEB), a diagnostic tool for assessing reasoning over internalized knowledge. Our evaluation on PEB revealed a significant gap: under the interactive paradigm of this benchmark, certain open-source reasoning models demonstrate strategic failures when attempting to apply procedural reasoning skills to specialized knowledge domains. This key finding, supported by our detailed error analyses, highlights a fundamental weakness in current LLMs and provides the community with a scalable, open-source construction method to guide and measure progress in closing this gap.

## 7 ETHICS STATEMENT

This work was conducted in adherence to the ICLR Code of Ethics. Our core research on model evaluation did not involve human subjects, live animals, or the collection of new personally identifiable information (PII).Our benchmarks are constructed from publicly available data. The PEB utilizes the English Wikipedia dump (2023-11-01), while the IC Back-End Puzzle Benchmark uses commands sourced from the official public documentation of EDA tools (Synopsys ICC/ICC2, Cadence Innovus). All data was used in compliance with the respective platforms' terms.To validate the difficulty of our benchmark, we recruited a small group of human annotators (PhD candidates). Participants were recruited through academic mailing lists and personal networks within relevant departments. Participation was voluntary, and all annotators were compensated at a rate exceeding the local minimum wage for their time. No sensitive personal data was collected during this verification process, and the annotation task did not involve exposure to harmful or offensive content.While we recognize that large-scale corpora like Wikipedia may reflect existing societal biases, our data curation and evaluation process were designed to be objective and not to amplify such biases. All models were accessed through their standard APIs, following their intended use guidelines.

# 8    REPRODUCIBILITY STATEMENT

We have made every effort to ensure the results in this paper are fully reproducible. To facilitate this, we have released a comprehensive supplementary package containing the complete source code and all datasets used in our experiments.

The provided code includes the entire Puzzle Everything benchmark system, from the core evaluation logic and API clients to the configuration files and experiment scripts. The datasets include the full Puzzle Everything Benchmark (PEB) and the IC Back-End Puzzle Benchmark used in our case study. Furthermore, our repository includes a detailed README file with instructions on prerequisites, installation, API configuration, and a minimal working example to validate the setup. The benchmarks themselves are derived from publicly accessible sources (the English Wikipedia dump and official EDA tool documentation), which are cited in the main text. We are confident that these resources will allow other researchers to replicate our experiments, verify our findings, and build upon our work.

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

APPENDIX

## A  THE USE OF LARGE LANGUAGE MODELS (LLMS)

In the preparation of this paper, LLMs and AI-powered tools were used for specific assistive tasks. The author is responsible for all core intellectual contributions. The roles of AI assistance are detailed below:

- Manuscript Preparation: The entire paper was written by the authors. LLMs were used as a tool for translation and to refine sentence structure and grammar for clarity.

- Code Development: The core experimental and analysis code was designed and written by the author. The AI tool Cursor was used to assist in optimizing and improving the architecture of certain parts of the code.

## B  RAPID CUSTOM BENCHMARK CONSTRUCTION

Our framework provides a scalable and generalizable method for rapidly constructing diagnostic benchmarks in any specialized domain. The process is straightforward and consists of two main stages:

- **Target Identification and Curation**: First, a domain of interest and a corresponding source corpus of knowledge are identified (e.g., technical manuals, API documentation, academic articles). From this corpus, a list of key concepts, terms, commands, or entities is systematically extracted to serve as the puzzle targets ($P_{\text{target}}$)

- **Reference Document Generation**: For each puzzle target, a "golden" reference document ($I_{\text{ref}}$) is established (e.g., the specific documentation page for a command). This document serves as the ground truth for the environment's feedback during the multi-turn evaluation.

### B.1  CASE STUDY: THE IC BACK-END PUZZLE BENCHMARK

To demonstrate the efficacy of this method, we applied it to the niche, industrial domain of IC back-end design. We used the official documentation for four widely-used EDA command setsSynopsys ICC, ICC2, and Cadence Innovus (both common and legacy)as our source corpus.

Table 3: Success rates (%) on the IC Back-End Puzzle Benchmark. The benchmark is composed of four command sets from specialized EDA tools. The columns correspond to Synopsys ICC, ICC2, and Cadence Innovus (**Invo. (c)** for common and **Invo. (le)** for legacy commands). The best performance in each domain and the best overall average are highlighted in **bold**.

| Model | ICC | ICC2 | Invo. (c) | Invo. (le) | Avg. |
|-------|-----|------|-----------|------------|------|
| Claude Opus 4 | **27.5** | 14.2 | 16.7 | 22.5 | **20.2** |
| GPT-5 | 25.8 | **15.0** | 14.2 | 20.0 | 18.8 |
| grok-4 | 15.0 | 10.8 | **18.3** | **24.2** | 17.1 |
| o3 | 20.0 | **15.0** | 12.5 | 12.5 | 15.0 |
| Claude Sonnet 4 | 14.2 | 10.8 | 9.2 | 18.3 | 13.1 |
| Gemini 2.5 Pro | 10.8 | 8.3 | 5.0 | 16.7 | 10.2 |
| Gemini 2.5 Pro No-Thinking | 10.8 | 8.3 | 6.7 | 12.5 | 9.6 |
| GPT-4o | 6.7 | 7.5 | 7.5 | 7.5 | 7.3 |
| DeepSeek V3 | 10.8 | 4.2 | 6.7 | 5.8 | 6.9 |
| GPT-OSS-120B | 8.3 | 7.5 | 4.2 | 4.2 | 6.0 |
| Qwen3 235B Instruct | 4.2 | 5.8 | 3.3 | 4.2 | 4.4 |
| Qwen3 235B Thinking | 6.7 | 2.5 | 3.3 | 5.0 | 4.4 |
| DeepSeek R1 | 4.2 | 0.8 | 2.5 | 2.5 | 2.5 |
| **Domain Avg.** | 12.7 | 8.5 | 8.5 | 12.0 | **10.4** |

Following our process, we curated 480 puzzles by sampling 120 command names from each of the four sets. This benchmark exemplifies how our framework can be used to quickly create robust, expert-level evaluations that test practical knowledge in any specialized domain.

We evaluated 13 leading LLMs on this benchmark, with the results presented in Table 3. The findings underscore the benchmark's significant challenge, with an average success rate of only 10.4% across all models. A clear performance hierarchy emerged, with Claude Opus 4 (20.2%), GPT-5 (18.8%), and Grok-4 (17.1%) forming a distinct top tier. Notably, reasoning-enhanced models like DeepSeek R1 (2.5%) and Qwen3 235B Thinking (4.4%) performed very poorly, reinforcing our main paper's finding of a gap between procedural reasoning and applied domain knowledge. The domain-specific results also reveal non-uniform knowledge; for instance, Grok-4 excels on Innovus commands but is weaker on ICC, suggesting that model knowledge of these specialized tools is often fragmented. This case study validates our method's ability to create challenging benchmarks that effectively differentiate model capabilities in highly specialized, real-world domains.

### B.2 BROADER APPLICATIONS AND FUTURE DIRECTIONS

The methodology detailed in this section is designed for broad applicability beyond the technical domains demonstrated. We encourage the research community to leverage our scalable process to construct new benchmarks across a wide array of fields. For example, puzzle targets could be drawn from the humanities (e.g., famous figures, artworks, film titles) or the natural sciences (e.g., celestial bodies, materials, biological species).

The creation of such diverse benchmarks would facilitate a more holistic evaluation of an LLM's all-around knowledge reasoning abilities. Moreover, the resulting multi-turn interaction data would provide a rich resource for deeper, cross-domain pattern recognition, enabling a more comprehensive exploration of model reasoning strategies and behaviors.

## C ENVIRONMENT AGENT ABLATION

To validate the robustness of our evaluation framework, we conducted a two-phase ablation study on the choice of the Environment Agent. The purpose of this study is to ensure that the observed performance of the strategic agents is not overly sensitive to the specific model used as the environment.

### C.1 PHASE 1: SUBSET ANALYSIS ACROSS MULTIPLE JUDGES

For our initial ablation studies, we created a representative 120-puzzle subset by randomly sampling 15 puzzles from each of the eight domains. This approach was chosen to ensure the reliability of the study while managing the significant computational costs of re-evaluation. All other experimental settings, such as MAX_ROUNDS and temperature, remained consistent with the main experiments.

As shown in Figure 6, some performance fluctuations are observable between the different environment agents, which is expected given the sampled dataset. However, despite these minor variations, each model's performance consistently falls within its established tier from the main experiments, and the overall performance trend is highly consistent across all three environments.

### C.2 PHASE 2: FULL-SCALE CROSS-VALIDATION WITH CLAUDE SONNET 4

To further verify the stability of the environment agent verification, we employed Claude Sonnet 4 to cross-validate the main results against DeepSeek V3, comprehensively re-evaluating the core benchmark (all 960 puzzles) across seven representative models.

**Rank Stability Across Different Judges**  As shown in Table 4, while absolute success rates increased under Claude Sonnet 4, the relative performance ordering remained remarkably stable:

- **Top-tier models:** GPT-5 and Claude Opus 4 maintain clear leadership in both environments.
- **Bottom-tier models:** "Reasoning-enhanced" models (DeepSeek R1, GPT-OSS-120B) remain at the bottom. Notably, even with Claude Sonnet 4 as the judge, DeepSeek R1 achieves only 8.6%

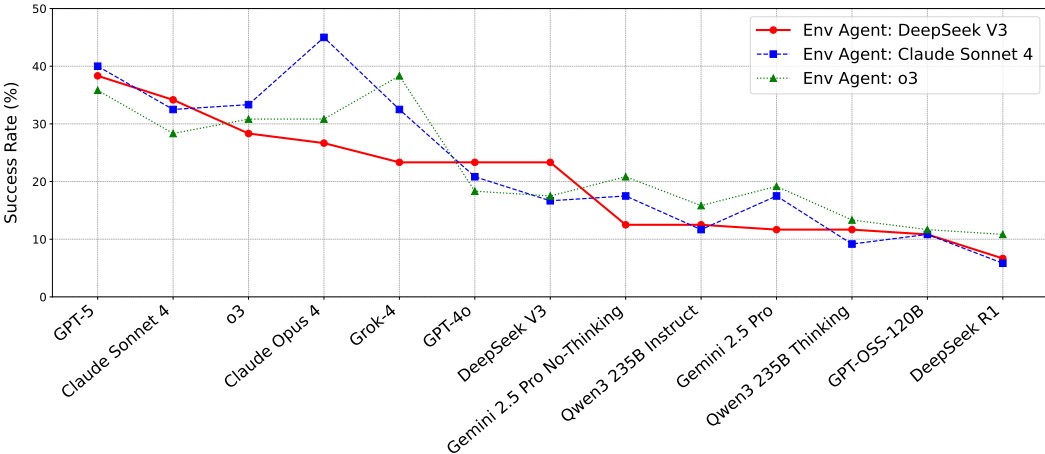

Figure 6: Ablation study on the Environment Agent. Success rates of 13 strategic agents are compared across three different environment agents (DeepSeek V3, Claude Sonnet 4, o3) on a 120-puzzle subset of PEB. The plot shows high consistency in the relative performance hierarchy of the models, validating the robustness of our framework and the choice of DeepSeek V3 as a cost-effective environment agent.

accuracy, which is far below standard models like DeepSeek V3 (17.4%) or GPT-4o (20.5%). This confirms that reasoning model failures stem from their internal strategies (i.e., the concrete-to-abstract fallacy), not judge-induced noise.

- **Correlation analysis:** The Spearman rank correlation between the two judges' average scores is $\rho = 0.96$. When comparing each model's 8-domain rankings across judges, the correlation remains high at $\rho = 0.89$.

Table 4: Comparison of Success Rates (%) using Claude Sonnet 4 vs. DeepSeek V3 as the Environment Agent on the full 960-puzzle benchmark. Bold values indicate the best performance in each domain under the Sonnet 4 judge. Comparison columns show the high stability of ranking despite absolute score shifts.

| Model | Bio | Dis | Chem | Phys | Math | CS | Econ | Phil | Avg.(Sonnet) | Avg.(V3) | Change |
|---|---|---|---|---|---|---|---|---|---|---|---|
| GPT-5 | **38.3** | 45.8 | **35.0** | **42.5** | **31.7** | 27.5 | **45.8** | 29.2 | **37.0** | 33.4 | - |
| Claude Opus 4 | 32.5 | **48.3** | 27.5 | 39.2 | 20.0 | **31.7** | **45.8** | **30.0** | 34.4 | 27.6 | - |
| Grok-4 | 26.7 | 43.3 | 30.0 | 26.7 | 17.5 | 22.5 | 36.7 | 29.2 | 29.1 | 17.6 | ↑ |
| GPT-4o | 20.8 | 33.3 | 18.3 | 16.7 | 9.2 | 14.2 | 31.7 | 20.0 | 20.5 | 19.6 | ↓ |
| DeepSeek V3 | 15.8 | 27.5 | 14.2 | 20.0 | 15.8 | 10.8 | 23.3 | 11.7 | 17.4 | 16.4 | - |
| GPT-OSS-120B | 13.3 | 20.8 | 12.5 | 12.5 | 8.3 | 9.2 | 15.0 | 10.8 | 12.8 | 13.4 | - |
| DeepSeek R1 | 5.0 | 20.8 | 5.0 | 10.0 | 5.0 | 5.0 | 10.8 | 7.5 | 8.6 | 7.2 | - |

**Explanation for Spikes and Judge Differences**    Our case analysis identified two primary factors driving the observed differences between judges: context robustness and semantic validation style.

- **Grok-4's outlier performance (Context Robustness):** As shown in Table 2 of the main paper, Grok-4 exhibits an anomalously high probing length (1930.48 characters/turn). DeepSeek V3 struggles to maintain precise instruction-following in such verbose contexts. In contrast, Claude Sonnet 4 effectively filters through Grok-4's lengthy outputs to extract core logic. This explains why Grok-4 shows the most dramatic improvement (+11.5%) under the stronger judge.

- **Judging characteristics (Association vs. Precision):** In our case analysis, we observed that DeepSeek V3 tends toward *associative validation*—occasionally accepting highly related "attributes" as the target "entity" itself, or applying looser boundaries for Boolean distinctions. In contrast, Claude Sonnet 4 enforces stricter semantic precision.

While V3's leniency introduces noise, our data demonstrate this noise affects most models uniformly and predictably at scale, rather than chaotically. Crucially, replacing DeepSeek V3 with Claude Sonnet 4 serves as a **linear scaling factor**. It uniformly "lifts" the scores of models capable of precise reasoning but does not rescue the fundamentally flawed strategies of the bottom-tier models. The high rank correlation confirms that DeepSeek V3 remains a valid, cost-effective proxy for differentiating model tiers.

## D    THEORETICAL PERSPECTIVE: PUZZLE SOLVING AS KNOWLEDGE GRAPH SEARCH

To provide a theoretical lens for interpreting the strategic behaviors observed in our experiments, we model the PEB task as a search problem over a latent Knowledge Graph. This perspective offers an information-theoretic explanation for why the "Concrete-to-Abstract" strategy observed in reasoning models results in inefficient convergence.

### D.1    INFORMATION GAIN AND SEARCH EFFICIENCY

We recall the definitions from the main text: the domain is modeled as a graph $G = (V, E)$ and the strategic agent seeks to identify $P_{\text{target}}$ within a limited budget $T$. At time step $t$, $S_t \subseteq V$ is the current set of candidate concepts.

An optimal agent seeks to generate a query $q_t$ that maximizes the expected reduction in uncertainty (entropy) over the candidate set $S_t$. Formally, this equates to maximizing Information Gain (IG):

$$q_t^* = \operatorname*{argmax}_q \left( H(S_t) - \mathbb{E}_a[H(S_t|q, a)] \right) \tag{1}$$

In the PEB context, high-IG queries typically target high-level attributes that act as "cutset" edges in the graph. By partitioning $S_t$ into semantically distinct clusters, such a strategy effectively prunes vast subgraphs. Under the ideal assumption that the ontology allows for relatively balanced partitioning, this "Coarse-to-Fine" approach approximates the efficiency of a tree traversal, where the expected search depth scales with the height of the taxonomy (often logarithmic relative to $|V|$), rather than the total number of entities.

### D.2    ANALYSIS OF FAILURE MODES

The "Concrete-to-Abstract" fallacy observed in reasoning models can be modeled as a sampling bias. Instead of selecting attributes that partition the graph, these models assign disproportionately high prior probabilities to specific leaf nodes based on weak associations. They then generate queries $q_t$ specifically designed to verify these sparse hypotheses.

From a search perspective, this strategy resembles random sampling or localized linear search. Without exploiting the hierarchical structure to disqualify large clusters of candidates simultaneously, the efficiency of this approach degrades significantly. In a worst-case scenario where the agent enumerates specific entities, the search complexity scales linearly with the size of the candidate pool ($O(|V|)$), making success statistically improbable within the strictly limited round budget (`MAX_ROUNDS=15`).

## E    TOKEN USAGE

This section details the token consumption patterns, averaged per puzzle, for both the strategic agent (SA) and the environment agent (EA). It is important to note that a single puzzle evaluation consists of multiple rounds of interaction..

The token cost of the environment agent (EA) is characterized by highly efficient outputs but demanding inputs. The EA's completion tokens are minimal and stable across all tests (averaging 460 tokens), as its responses are restricted to a fixed format providing only correctness judgments. This ensures a fair and consistent evaluation. However, its prompt token requirements are substantial, averaging 18,347 tokens per puzzle, because the EA must process the full reference document ($I_{\text{ref}}$) in each round to provide accurate feedback.

Similarly, the strategic agents (SA) prompt costs are significant, averaging 18,826 tokens per puzzle, as the full interaction history is included in the context for each new turn. The vast majority of performance variance and economic cost, however, comes from the SAs completion tokens, which reflect model verbosity (Table 5). Notably, some of the poorest-performing open-source reasoning models were paradoxically the most computationally expensive. GPT-OSS-120B was the most costly, consuming over 72,000 total SA tokens on average, followed by Qwen3 235B Thinking (approx. 57,600 tokens) and DeepSeek R1 (approx. 43,300 tokens). Collectively, these three open-source models averaged 57,784 SA tokens per puzzlealmost exactly double the 28,705-token average of the remaining models. This contrasts sharply with top-performing models like GPT-5, which achieved the highest success rate while using less than half the tokens of GPT-OSS-120B. This suggests that excessive verbosity is not only an unsuccessful strategy but also an economically inefficient one for this task.

Table 5: A summary of token usage for representative models, averaged per puzzle. "SA Tokens" refers to the strategic agent. "EA Completion Tokens" refers to the environment agent's output, which is consistently low due to a fixed response format.

| Model | SA Prompt | SA Completion | SA Total Tokens | EA Completion Tokens |
|---|---|---|---|---|
| *(Best Performance)* | | | | |
| GPT-5 | 9,821 | 24,301 | 34,122 | 430 |
| *(Most Verbosity)* | | | | |
| GPT-OSS-120B | 14,034 | 58,471 | 72,505 | 473 |
| Qwen3 235B Thinking | 33,414 | 24,170 | 57,584 | 473 |
| Deepseek R1 | 15,242 | 28,021 | 43,263 | 458 |
| *(Most Concise)* | | | | |
| GPT-4o | 13,441 | 1,513 | 14,954 | 466 |
| **Average (all models)** | 18,826 | 16,583 | 35,416 | 460 |

# F   PUZZLE EVERYTHING PROMPT

---

**Strategic Agent Prompt**

You are a `{domain}` expert. I will have a preset `{domain}` name/term/concept in mind, and you need to continuously ask questions and reason until you can identify the name/term/-concept. You can ask questions to narrow down the possibilities. Your questions must be in the form of a single, four-option multiple-choice question at a time.

You can only ask one multiple-choice question at a time. Each question is crucial and should be based on your reasoning, professional knowledge, and the previous rounds of questions and answers. Please try to identify the term/concept in as few rounds as possible.

I have now preset a name/term/concept. Please begin with your first question.

---

Figure 7: The system prompt is for the strategic agent. In each round, it poses strategic queries to quickly identify the $P_{\text{target}}$ by interacting with an environment. The `{domain}` placeholder is replaced with the specific domain name (e.g., Human Disease, Physics) for each evaluation.

---

**Environment Agent Prompt**

You are a `{domain}` expert. You will have a preset `{domain}` name/term/concept in mind, and I need to identify it by asking questions and using reasoning. I can ask questions to narrow down the scope, and my method of questioning will be a single, four-option multiple-choice question at a time. The specific details regarding my question are all within the <qa_response> part. You will respond to my questions based on some "golden information" (reference material).

The name/term/concept you have preset is: `{puzzle_name}`
Reference material: <reference_material> `{puzzle_reference}`</reference_material>

In your reply, you must include a <pure_parse>...</pure_parse> block and a <pure_answer>...</pure_answer> block.

**For the `<pure_parse>...</pure_parse>` block:**
You will only judge the correctness of the options. For example, when option A, D are correct, and option B, C are incorrect: <pure_parse>A is correct, B is incorrect, C is incorrect, D is correct</pure_parse>
When all options are incorrect, reply with <pure_parse>A is incorrect, B is incorrect, C is incorrect, D is incorrect</pure_parse>
If the <qa_response> part provides only an incorrect final guess without including a multiple-choice question, reply with <pure_parse>Your guess is incorrect, please ask a new question</pure_parse> and <pure_answer>None</pure_answer>. The <pure_parse> block is forbidden from containing the name of the preset name/term/concept or any other content from the <reference_material> part.

**For the `<pure_answer>...</pure_answer>` block:**
If the message you receive in the <qa_response> part (or one of the options in the <qa_response> part) explicitly states the name of your preset name/term/concept (or its alias), then include <pure_answer>Your preset name</pure_answer> in your reply (Here, "Your preset name" is replaced with the actual name that appears). If it does not, then include <pure_answer>None</pure_answer>.

All of your replies must include both of these blocks.

---

Figure 8: The system prompt for the environment agent. This agent acts as an oracle, providing structured and non-leaking feedback to the strategic agent based on the $P_{\text{target}}$ and $I_{\text{ref}}$.

## G DATA CONSTRUCTION DETAILS AND DIFFICULTY VALIDATION

In this section, we provide a more granular breakdown of the data construction pipeline and offer objective validation for the "doctoral-level" difficulty of the selected puzzles.

### G.1 CONSTRUCTION PIPELINE

Our construction process employs a rigorous, multi-stage distillation pipeline designed to isolate highly specialized concepts from a raw dump of 6.4 million English Wikipedia titles. The pipeline retained only approximately 9,000 final targets, resulting in a strictly controlled **0.14% retention rate**. The process is summarized in four distinct stages:

1. **Stage 1 (Broad Filter):** We performed a rapid removal of common entities, pop culture references, and general knowledge terms to form an initial candidate pool.

2. **Stage 2 (Doctoral Criteria):** We applied explicit prompt-based filtering requirements, instructing the model to retain only titles requiring "doctoral-level specialized knowledge" while discarding undergraduate or introductory concepts.

3. **Stage 3 (Domain Stratification):** The remaining specialized candidates were rigorously categorized into the eight designated academic fields (Biology, Physics, etc.) to ensure balanced domain representation.

4. **Stage 4 (Dynamic Difficulty Selection):** This stage served as a dynamic tournament selection. From a candidate pool of approximately 150 specialized terms per batch, we dynamically selected only the top 5–10 most challenging items based on model difficulty scoring. This ensures that the final puzzles represent the "tail of the tail" in terms of difficulty distribution.

To facilitate reproducibility and further research, the complete filtering code and associated prompts are openly available.[1]

### G.2 VALIDATION OF DIFFICULTY

To validate the difficulty of the benchmark, we employed both objective topological analysis and human verification.

**Topological Analysis (Knowledge Graph Structure)**

To objectively quantify "difficulty" without relying solely on LLM subjectivity, we conducted a topological analysis based on Wikipedia's link structure. We measured the **In-link Count** (a proxy for PageRank centrality) for entities at different stages of our filtering pipeline. In knowledge graphs, high in-link counts typically indicate common, foundational concepts (central nodes), while low in-link counts indicate specialized, niche concepts (edge nodes).

As shown in Table 6, the Median In-link count drops dramatically from over 500 in the rejected group to just 16 in our final selection. The fact that Group 1 hit the API saturation limit confirms they are central "knowledge hubs," whereas the PEB concepts are distinct "leaf nodes." This demonstrates that the PEB successfully targets the long-tail, edge nodes of the knowledge graph, objectively corresponding to high specialization.

Table 6: Topological Analysis of Entity Difficulty via In-link Counts. The drastic drop in median in-links confirms that the final selection consists of highly specialized "edge" knowledge.

| Group | Sample Size | Mean | Median | Implication |
|---|---|---|---|---|
| G1: Rejected at Stage 1 | 1000 | 379.22 | 500+* | Common Knowledge |
| G2: Kept at S2, Rejected at S4 | 1000 | 86.72 | 19.00 | Specialized Knowledge |
| **G3: Final Selection (PEB)** | **1000** | **59.30** | **16.00** | **Deep/Edge Knowledge** |

*Note: The Wikipedia API caps single-request backlink counts at 500. Group 1 hitting this ceiling indicates saturation.

---

[1] https://anonymous.4open.science/r/wikipedia_filter-7870

**Human Verification** Complementing this objective metric, we recruited Ph.D. candidates from relevant disciplines to manually audit a random sample of 400 puzzles (50 per domain). The human experts corroborated the "doctoral-level" classification, confirming that the targets required deep domain expertise not typically found in general undergraduate curricula.

# H BENCHMARK COMPOSITION

Table 7: Biology Domain Composition (120 total entries)

| Sub-Discipline | Entries | Percentage |
|---|---|---|
| **Molecular and Genomic Biology** | **29** | **24.2%** |
| Genes, Proteins, and Nucleic Acids | 16 | 13.3% |
| Gene Expression and Regulation | 7 | 5.8% |
| DNA Structure and Dynamics | 5 | 4.2% |
| Genetics and Inheritance | 1 | 0.8% |
| **Ecology and Evolution** | **24** | **20.0%** |
| Medical Biology and Pathology | 10 | 8.3% |
| Evolutionary Biology and Genetics | 5 | 4.2% |
| Biophysics and Theoretical Biology | 5 | 4.2% |
| Ecology and Environmental Science | 4 | 3.3% |
| **Biotechnology and Bioengineering** | **20** | **16.7%** |
| Molecular and Genetic Techniques | 9 | 7.5% |
| Other Techniques and Models | 5 | 4.2% |
| Imaging and Microscopy | 3 | 2.5% |
| Drug Discovery and Delivery | 3 | 2.5% |
| **Systems and Omics Biology** | **12** | **10.0%** |
| Systems-Level Concepts and Models | 8 | 6.7% |
| "-omics" Fields | 4 | 3.3% |
| **Cellular and Subcellular Biology** | **10** | **8.3%** |
| Cellular Processes | 3 | 2.5% |
| Cell Death | 3 | 2.5% |
| Organelles and Subcellular Structures | 2 | 1.7% |
| Cytoskeleton and Molecular Motors | 2 | 1.7% |
| **Biochemistry and Metabolism** | **10** | **8.3%** |
| Proteins and Enzymes | 8 | 6.7% |
| Metabolic Pathways | 2 | 1.7% |
| **Immunology** | **9** | **7.5%** |
| Core Concepts and Processes | 5 | 4.2% |
| Signaling and Receptors | 4 | 3.3% |
| **Neuroscience** | **6** | **5.0%** |
| Molecular and Cellular Neuroscience | 2 | 1.7% |
| Systems and Cognitive Neuroscience | 2 | 1.7% |
| Neuropharmacology | 1 | 0.8% |
| Clinical Neuroscience | 1 | 0.8% |

Table 8: Human Disease Domain Composition (120 total entries)

| Sub-Discipline | Entries | Percentage |
|---|---|---|
| **Systemic & Structural Genetic Disorders** | **35** | **29.2%** |
| Multi-system Genetic Syndromes | 14 | 11.7% |
| Skeletal and Overgrowth Syndromes | 13 | 10.8% |
| Syndromes with Major Dermatologic Features | 4 | 3.3% |
| Genomic and Inheritance Disorders | 4 | 3.3% |
| **Neurogenetic Disorders** | **18** | **15.0%** |
| Neurodevelopmental Syndromes | 13 | 10.8% |
| Hereditary Neuromuscular and Neurodegenerative Diseases | 5 | 4.2% |
| **Oncology** | **15** | **12.5%** |
| Hematologic Malignancies | 6 | 5.0% |
| Solid Tumors | 6 | 5.0% |
| Hereditary Cancer Syndromes | 2 | 1.7% |
| Transmissible Cancers | 1 | 0.8% |
| **Neurology** | **14** | **11.7%** |
| Neurodegenerative and Movement Disorders | 4 | 3.3% |
| Epilepsy and Seizure Disorders | 3 | 2.5% |
| Other Neurological Conditions | 3 | 2.5% |
| Neuroinflammatory and Demyelinating Diseases | 2 | 1.7% |
| Congenital and Structural Brain Disorders | 2 | 1.7% |
| **Immuno & Rheum** | **14** | **11.7%** |
| Autoinflammatory and Autoimmune Syndromes | 7 | 5.8% |
| Systemic Vasculitides | 4 | 3.3% |
| Primary Immunodeficiencies | 2 | 1.7% |
| Systemic and Catastrophic Conditions | 1 | 0.8% |
| **Endo & Metab** | **13** | **10.8%** |
| Inborn Errors of Metabolism | 11 | 9.2% |
| Disorders of Mineral and Porphyrin Metabolism | 2 | 1.7% |
| **Organ-System** | **11** | **9.2%** |
| Hematologic Disorders | 4 | 3.3% |
| Gastroenterology and Hepatology | 3 | 2.5% |
| Dermatologic Disorders | 1 | 0.8% |
| Pulmonology | 1 | 0.8% |
| Infectious and Transmissible Diseases | 1 | 0.8% |
| Other Conditions | 1 | 0.8% |

Table 9: Chemistry Domain Composition (120 total entries)

| Sub-Discipline | Entries | Percentage |
|---|---|---|
| **Organic Chemistry** | **27** | **22.5%** |
| Named Reactions and Rearrangements | 11 | 9.2% |
| Synthesis, Reagents, and Specific Compounds | 8 | 6.7% |
| Reaction Mechanisms and Concepts | 5 | 4.2% |
| Stereochemistry and Molecular Structure | 3 | 2.5% |
| **Biochemistry and Chemical Biology** | **27** | **22.5%** |
| Enzymes and Cofactors | 11 | 9.2% |
| Biological Processes and Pathways | 8 | 6.7% |
| Bio-analytical and Chemical Biology Tools | 6 | 5.0% |
| Bio-inorganic Chemistry | 2 | 1.7% |
| **Physical Chemistry** | **19** | **15.8%** |
| Quantum Chemistry and Spectroscopy | 6 | 5.0% |
| Electrochemistry | 4 | 3.3% |
| Thermodynamics and Calorimetry | 3 | 2.5% |
| Kinetics and Dynamics | 3 | 2.5% |
| Condensed Matter and Surface Chemistry | 3 | 2.5% |
| **Inorganometallics and Organometallics** | **17** | **14.2%** |
| Organometallic Compounds | 8 | 6.7% |
| Specific Inorganic Compounds | 6 | 5.0% |
| Bonding and Molecular Structure | 3 | 2.5% |
| **Analytical Chemistry** | **11** | **9.2%** |
| Spectroscopy and Spectrometry | 5 | 4.2% |
| Instrumentation and Databases | 3 | 2.5% |
| Microscopy and Surface Analysis | 2 | 1.7% |
| Chromatography and Separation | 1 | 0.8% |
| **Applied Chemistry** | **8** | **6.7%** |
| Photochemistry and Photophysics | 2 | 1.7% |
| Magnetochemistry | 2 | 1.7% |
| Industrial and Applied Chemistry | 2 | 1.7% |
| Radiochemistry and Exotic Matter | 1 | 0.8% |
| Theoretical Physics in Chemistry | 1 | 0.8% |
| **Materials Chemistry** | **6** | **5.0%** |
| Key Concepts and Materials | 6 | 5.0% |
| **Theoretical Chemistry** | **5** | **4.2%** |
| Concepts and Models | 3 | 2.5% |
| Methods and Techniques | 2 | 1.7% |

Table 10: Physics Domain Composition (120 total entries)

| Sub-Discipline | Entries | Percentage |
|---|---|---|
| **Quantum Field Theory** | **22** | **18.3%** |
| Foundational and Axiomatic QFT | 9 | 7.5% |
| Gauge Theories and the Standard Model | 6 | 5.0% |
| Non-Perturbative and Topological QFT | 3 | 2.5% |
| Supersymmetry and Beyond | 2 | 1.7% |
| QFT in Curved Spacetime | 2 | 1.7% |
| **Mathematical Physics** | **19** | **15.8%** |
| Differential Geometry and Topology | 6 | 5.0% |
| Group Theory and Algebra | 5 | 4.2% |
| Functional Analysis and Operator Theory | 5 | 4.2% |
| Integrable Systems and Exact Solutions | 2 | 1.7% |
| Twistor Theory | 1 | 0.8% |
| **Quantum Mechanics** | **16** | **13.3%** |
| Foundational Concepts and Formalism | 6 | 5.0% |
| Quantum Systems and Dynamics | 4 | 3.3% |
| Quantum Computation and Information | 4 | 3.3% |
| Quantum States and Phenomena | 2 | 1.7% |
| **General Relativity** | **16** | **13.3%** |
| Alternative and Extended Theories of Gravity | 7 | 5.8% |
| Formalisms and Solutions | 5 | 4.2% |
| Astrophysical Relativity | 3 | 2.5% |
| Spacetime and Black Holes | 1 | 0.8% |
| **Specialized Topics in Physics** | **16** | **13.3%** |
| Atomic, Molecular, and Optical (AMO) Physics | 7 | 5.8% |
| Particle and Nuclear Physics | 5 | 4.2% |
| Plasma Physics | 4 | 3.3% |
| **Condensed Matter** | **14** | **11.7%** |
| Quantum Materials and Phases | 6 | 5.0% |
| Electronic and Spintronic Effects | 4 | 3.3% |
| Many-Body and Superconducting Systems | 2 | 1.7% |
| Statistical and Soft Matter | 2 | 1.7% |
| **String Theory** | **12** | **10.0%** |
| Core Concepts and Dualities | 6 | 5.0% |
| Advanced and Related Theories | 5 | 4.2% |
| Cosmological Applications | 1 | 0.8% |
| **Classical Physics and Electrodynamics** | **5** | **4.2%** |
| Core Concepts and Phenomena | 5 | 4.2% |

Table 11: Mathematics Domain Composition (120 total entries)

| Sub-Discipline | Entries | Percentage |
|---|---|---|
| **Geometry & Topology** | **37** | **30.8%** |
| Algebraic Topology & Homotopy Theory | 12 | 10.0% |
| General & Geometric Topology | 12 | 10.0% |
| Differential & Geometric Analysis | 11 | 9.2% |
| K-Theory | 2 | 1.7% |
| **Number Theory** | **19** | **15.8%** |
| Arithmetic & Algebraic Geometry | 12 | 10.0% |
| Algebraic Number Theory & Class Field Theory | 5 | 4.2% |
| Analytic Number Theory | 2 | 1.7% |
| **Analysis** | **18** | **15.0%** |
| Operator Theory & Operator Algebras | 9 | 7.5% |
| Real, Complex, & Harmonic Analysis | 8 | 6.7% |
| Control Theory | 1 | 0.8% |
| **Cryptography & Logic** | **18** | **15.0%** |
| Cryptography | 7 | 5.8% |
| Theoretical Computer Science & Algorithms | 6 | 5.0% |
| Logic & Set Theory | 5 | 4.2% |
| **Algebra & Group** | **14** | **11.7%** |
| Homological & Abstract Algebra | 8 | 6.7% |
| Representation Theory | 4 | 3.3% |
| Group Theory | 2 | 1.7% |
| **Theo Phys** | **8** | **6.7%** |
| Quantum Mechanics & Quantum Field Theory | 3 | 2.5% |
| General Relativity & Spacetime Physics | 2 | 1.7% |
| Plasma & Fluid Dynamics | 1 | 0.8% |
| Physical Chemistry | 1 | 0.8% |
| Dynamical Systems | 1 | 0.8% |
| **Statistics** | **6** | **5.0%** |
| Probability & Statistics | 4 | 3.3% |
| Stochastic Analysis | 2 | 1.7% |

Table 12: Computer Science Domain Composition (120 total entries)

| Sub-Discipline | Entries | Percentage |
|---|---|---|
| **Theoretical Computer Science** | **43** | **35.8%** |
| Mathematical Foundations | 25 | 20.8% |
| Logic in Computer Science | 12 | 10.0% |
| Computability and Complexity Theory | 6 | 5.0% |
| **Artificial Intelligence and Machine Learning** | **17** | **14.2%** |
| Bayesian and Probabilistic Methods | 5 | 4.2% |
| Learning Theory and Models | 4 | 3.3% |
| Specialized Domains and Applications | 4 | 3.3% |
| Logic, Reasoning, and Knowledge Representation | 3 | 2.5% |
| Robotics and Embodied AI | 1 | 0.8% |
| **Programming Languages and Formal Methods** | **13** | **10.8%** |
| Formal Verification | 4 | 3.3% |
| Automata and Formal Languages | 4 | 3.3% |
| Type Theory and Program Analysis | 3 | 2.5% |
| Formal Semantics and Rewriting Systems | 2 | 1.7% |
| **Scientific Computing** | **13** | **10.8%** |
| Computational Physics and Engineering | 6 | 5.0% |
| Numerical Methods | 5 | 4.2% |
| Optimization and Control | 2 | 1.7% |
| **Cryptography** | **12** | **10.0%** |
| Cryptosystems and Primitives | 4 | 3.3% |
| Cryptanalysis | 4 | 3.3% |
| Secure Protocols | 2 | 1.7% |
| Foundational Concepts | 2 | 1.7% |
| **Algorithmic Theory** | **11** | **9.2%** |
| Graph and String Algorithms | 3 | 2.5% |
| Algorithmic Information Theory | 3 | 2.5% |
| Data Structures and Compression | 2 | 1.7% |
| Combinatorial Optimization | 2 | 1.7% |
| Algorithmic Game Theory | 1 | 0.8% |
| **Quantum Computing** | **11** | **9.2%** |
| Quantum Algorithms and Information | 4 | 3.3% |
| Quantum Foundations and Simulation | 4 | 3.3% |
| Quantum Complexity and Models | 2 | 1.7% |
| Interdisciplinary Applications | 1 | 0.8% |

Table 13: Economics Domain Composition (120 total entries)

| Sub-Discipline | Entries | Percentage |
|---|---|---|
| **Pluralist Econ** | **34** | **28.3%** |
| Behavioral and Experimental Economics | 8 | 6.7% |
| Public Economics and Social Choice Theory | 8 | 6.7% |
| Marxian Economics | 7 | 5.8% |
| Other Theories and Applied Fields | 6 | 5.0% |
| Economic History and History of Thought | 5 | 4.2% |
| **Microeconomics** | **19** | **15.8%** |
| Consumer and Firm Theory | 13 | 10.8% |
| General Equilibrium and Market Design | 6 | 5.0% |
| **Econometrics** | **15** | **12.5%** |
| Models and Methods | 11 | 9.2% |
| Statistical Concepts | 4 | 3.3% |
| **Macroeconomics** | **14** | **11.7%** |
| Economic Models and Growth | 8 | 6.7% |
| Monetary Policy and Critiques | 6 | 5.0% |
| **Math/Comp Econ** | **14** | **11.7%** |
| Models and Methods | 14 | 11.7% |
| **Game Theory** | **9** | **7.5%** |
| Concepts and Models | 9 | 7.5% |
| **Finance** | **9** | **7.5%** |
| Asset Pricing and Risk Management | 9 | 7.5% |
| **International Economics** | **6** | **5.0%** |
| Trade and Finance | 6 | 5.0% |

Table 14: Philosophy Domain Composition (120 total entries)

| Sub-Discipline | Entries | Percentage |
|---|---|---|
| **Philosophy of Religion and Interdisciplinary** | **31** | **25.8%** |
| History of Philosophy and Specific Texts | 10 | 8.3% |
| Philosophy of Religion and Theology | 9 | 7.5% |
| Other Interdisciplinary Concepts | 8 | 6.7% |
| Philosophy of Language | 4 | 3.3% |
| **Logic and Philosophy of Mathematics** | **20** | **16.7%** |
| Modal and Non-Classical Logics | 7 | 5.8% |
| Logic in Computation and Language | 7 | 5.8% |
| Set Theory and Mathematical Foundations | 6 | 5.0% |
| **Epistemology & Phil Sci** | **18** | **15.0%** |
| Epistemology | 10 | 8.3% |
| Philosophy of Science | 4 | 3.3% |
| Philosophy of Physics | 4 | 3.3% |
| **Metaphysics & Ontology** | **16** | **13.3%** |
| Core Concepts | 9 | 7.5% |
| Theories of Reality | 4 | 3.3% |
| Metaphysics of Mind | 3 | 2.5% |
| **Continental Philosophy** | **13** | **10.8%** |
| Phenomenology and Existentialism | 5 | 4.2% |
| Post-Structuralism and Deconstruction | 4 | 3.3% |
| Lacanian Psychoanalysis | 4 | 3.3% |
| **Eastern Philosophy** | **12** | **10.0%** |
| Indian Philosophy (Hindu Schools) | 6 | 5.0% |
| Indian Philosophy (Buddhist Schools) | 5 | 4.2% |
| Indian Religious/Philosophical Concepts | 1 | 0.8% |
| **Social Philosophy** | **6** | **5.0%** |
| Legal and Social Philosophy | 5 | 4.2% |
| Political Philosophy | 1 | 0.8% |
| **Mind** | **4** | **3.3%** |
| Core Concepts | 3 | 2.5% |
| Psychoanalytic Philosophy | 1 | 0.8% |

