# OpenReview forum: "Puzzle Everything: Benchmarking Knowledge Reasoning through Generalized Puzzle Solving"
_ICLR.cc/2026/Conference — Submitted to ICLR 2026_

### Official Review · Reviewer_Jwi6 · 2025-10-29

**Soundness:** 2
**Presentation:** 1
**Contribution:** 2
**Rating:** 2
**Confidence:** 4

**Summary:**

The paper presents the Puzzle Everything Benchmark (PEB), which is designed to test reasoning over *internal knowledge* by presenting the models with a specific form of puzzle. In each puzzle, there is a hidden piece of information, e.g. a name or a concept, and the model being tested must iteratively ask questions to figure out the information. Each round, it receives feedback from an *environment agent* which has access to the concept's Wikipedia page in-context. Successful identification of the concept requires asking sensible questions and using the returned information effectively. This benchmark tests models' abilities to navigate the search for unknown information and to update on new information. The methodology can be broadly applied to information from different domains.

**Strengths:**

- **Originality:** This paper presents an interesting and novel method for testing LLMs. The motivation for the paper is that many benchmarks test knowledge directly, which is a recall task. This becomes trivial if models have been trained on the information, which is very possible given the current paradigm of training. The benchmarking situation is an urgent problem demanding new solutions.
- **Comprehensiveness:** The benchmark is broad, containing 960 questions on 8 major topics. This is a relative strength. Similarly, it can be applied to any new piece of information. It reminds me a little bit of the capability being tested in [1], which might be an interesting reference.
- **Interesting metrics:** The analysis and error analysis sections are particularly interesting. The incorrect guessing metric is a real insight into the overconfidence of different models. This is a nice novel idea. Additionally, the CoT analysis to work out the strategies that models take is interesting (though it may be wise for the authors to hedge that CoT is not necessarily faithful here, e.g. [2]).
- **Reverse scaling laws:** One of the potentially highly impactful contributions of the paper is an example of negative returns to test-time compute. This would be a significant discovery if it were the case! However, this hypothesis needs to be explored further to be convincing (see *weaknesses* section).

[1] The reversal curse: LLMs trained on "A is B" fail to learn "B is A". Berglund et al. 2024.
[1] Chain-of-Thought Is Not Explainability: Barez et al. 2025.

**Weaknesses:**

- **Phenomenon:** It is unclear to me exactly what the specific phenomenon being tested is or how generalisable this is to other tasks. The authors present it in the title as *knowledge reasoning* and expand this to reasoning over *internal knowledge* throughout the paper. If the authors could add precise definitions of these terms and the phenomenon the benchmark targets, it would aid the paper. For example, one question I had was, what are examples of this skill outside of the specific problem domain? Furthermore, it is unclear whether better performance comes from better factual knowledge or better knowledge of good strategies for the specific game mechanics. This is noted as a limitation of *existing* puzzle benchmarks (L081), hence it would be nice if the paper made it clearer why this does not apply here.
- **Method explanation:** Key parts of the methodology are not properly explained in the main paper which made it hard to follow, e.g. How were the six million article titles screened to be doctoral-level difficulty or greater? How were they grouped into higher-level topics? Clearer explanation here would help.
- **Writing and presentation:** There could be some quick improvements on the writing and presentation. In places, the language is a bit imprecise (e.g. L142, L161) or excessively descriptive/not in standard academic paper style (e.g. L262). Similarly, some of the figures could be iterated on, e.g. Figure 4 is unclear - what does the shading mean, and what does the size of each sub-block mean (does this represent the proportion of questions belonging to that sub-topic?)
- **Reasoning vs non-reasoning:** The authors discover a potentially very interesting finding that reasoning models perform worse than their non-reasoning counterparts. This is cool! However, more experiments are needed to fully support this claim. For example, GPT-5 could be run on low, medium, and high thinking modes to see if returns to test time compute really are negative. I would be particularly interested in seeing the results of this. One potential reason for this might be that the reasoning models are being run at temperature 0.2. There is fairly good evidence that reasoning-style models perform worse at low temperatures [1], so it might be worth doing an ablation with each model's recommended inference parameters. In addition, I was under the impression that OpenAI don't let you set the temperature parameter for GPT-5 (I think for this reason), so I'm confused what parameters were used here.
- **Importance of the environment agent:** The environment agent seems to have a highly non-trivial task of accurately answering the questions. Even with the Wikipedia page in-context, I would imagine this is a hard task! It would be good to show some accuracy metric of the environment agent or a proxy of agreement between different models on the same tasks. I would expect a fair amount of variations, which complicates the benchmark. Similarly, the paper reports that the choice of environment agent is not important, yet Figure 6 in the appendix shows that 3 different environment agents would lead to 3 different best models on the benchmark. This seems very significant. Perhaps it would be best to average over different agents?
- **Lack of statistical analysis:** In a similar vein, the results are missing statistical analysis, like error bars, which makes comparison hard. For example, it is hard to compare the relative performance of the DeepSeek, Gemini and Qwen variants without this.

This is definitely an interesting benchmark that could be published with further revision; however, in my opinion, there are currently a few significant issues that mean it's not quite ready.

[1] Qwen 3 8B Huggingface: https://huggingface.co/Qwen/Qwen3-8B

**Questions:**

A few questions on things I didn't pick up from the paper (I may have missed these, so would appriciate help understanding them)

[1] **Inference settings:** As before, the paper says models are run on temp. 0.2. Are all models run on that temp?

[2] **CoT Access:** To what extent can models see their previous CoTs? With open source models this would be possible, with closed models, this is semi-possible, e.g. the Claude API has some functionality for this. I belive this is not possible with OpenAI.

[3] **Code:** Can you release the code in an anonymous repository (e.g. https://anonymous.4open.science)?

**Other points:**
In Figure 2, the model appears to misunderstand the task initially, starting its response: “I’ll help you identify the disease.” How common is this misunderstanding?

---

> ### Author Response · Authors · 2025-11-24
> **Author Response Part Ⅰ**
>
> Thank you for your comments and helpful suggestions. We address your concerns in detail below.
> > **Weakness in Phenomenon Being Tested**
>
> **1. What phenomenon does PEB test?**
> Our benchmark evaluates **ontology-aware knowledge reasoning** – the ability to navigate hierarchical conceptual structures through strategic inquiry. A concrete real-world analog is **medical diagnosis**: a physician identifies a preset disease through multi-turn questioning of a patient, systematically narrowing possibilities by probing symptoms, body systems, and pathophysiology. While our framework is domain-general rather than medicine-specific, the underlying cognitive capabilities are parallel.
>
> * **Theoretical grounding:** The puzzle-solving process is fundamentally an *ontological traversal task*. In **Appendix D**, we further provide a theoretical analysis perspective to help understand this. Strategic agents must:
>     1.  Infer the implicit taxonomic structure of the domain (parent-child relationships, class hierarchies).
>     2.  Formulate queries that efficiently partition this space by testing attributes and relationships.
>     3.  Progressively localize the target concept within the knowledge graph.
>     *This aligns with work on ontological knowledge in language models [1].*
>
> **2. Why this differs from "learning game mechanics" (L081)**
> The reviewer correctly notes that traditional puzzle benchmarks may conflate strategic gaming ability with domain reasoning. PEB addresses this in three ways:
>
> * **Domain knowledge is prerequisite:** Unlike Sudoku or chess where rules are self-contained, success requires internalized knowledge of the domain's conceptual structure. A model cannot solve a Human Disease puzzle through domain-agnostic strategies (e.g., character-by-character guessing is prohibited by our MCQ format).
> * **Strategies converge to domain-invariant patterns:** Our empirical analysis reveals that successful models employ remarkably consistent **coarse-to-fine** questioning strategies across all domains (Figure 2, Table 2). All top-performing models begin with broad categorical questions (e.g., "Which body system?" in Disease, "Which subfield?" in Physics) and progressively narrow scope through hierarchical refinement. This convergence indicates models are executing genuine deductive reasoning over knowledge structures, not memorizing domain-specific game tricks. Conversely, failed attempts (e.g., DeepSeek R1's "concrete-to-abstract" pattern in Figure 5) represent strategy failures, not lack of game mechanics knowledge.
> * **Implicit ontologies vary by domain:** While some domains (diseases, species, celestial bodies) have explicit taxonomies, others (e.g., Physics, Philosophy) have latent hierarchical structures. Performance reflects the joint requirement of (a) possessing specialized domain knowledge (the explicit target concept and its attributes) and (b) reasoning strategically over the implicit ontological structure. Strong game-playing without domain knowledge fails; domain knowledge without strategic reasoning is inefficient.
>
> [1] [Do PLMs Know and Understand Ontological Knowledge?](https://aclanthology.org/2023.acl-long.173/) (Wu et al., ACL 2023)
>
> > **Weakness in Method Explanation**
>
> **Please refer to General Response Q1.** We have provided a detailed breakdown of the **0.14% coarse-to-fine filtering pipeline** and the topological validation of "doctoral-level" difficulty.
>
> > **Weakness in Importance of the Environment Agent**
>
> **Please refer to General Response Q2.** We empirically validated that while the choice of Environment Agent affects absolute scores, the relative performance hierarchy (especially the underperformance of reasoning models) remains highly stable ($\rho > 0.96$) across different judges.

---

> ### Author Response · Authors · 2025-11-24
> **Author Response Part Ⅱ**
>
> > **Weakness in Lack of Statistical Analysis**
>
> We recognize the importance of statistical rigor in comparing model performance. However, it is crucial to note that distinct from single-turn QA benchmarks, PEB involves a **multi-turn interaction process** (averaging 14.02 rounds per puzzle, as shown in Table 2). This interactive nature elevates computational and temporal costs by an order of magnitude. Consequently, applying standard bootstrapping across the full suite of 13 models would incur prohibitive costs.
>
> **Alternative Robustness Verification (Sample Size Analysis):**
> To address stability concerns—particularly for comparisons between variants of DeepSeek, Gemini, and Qwen—without exponential resource expenditure, we validated our results by analyzing rank stability as a function of sample size and environment agent (please refer to **General Response Q2**).
>
> Our empirical analysis reveals a distinct **"Rank Convergence"** phenomenon, confirming that as the dataset size increases, the reported performance differences represent statistically robust signals rather than noise:
>
> * **Global Stability:** When expanding the evaluation set from the ablation subset ($N=120$) to the full benchmark ($N=960$), the Spearman rank correlation ($\rho$) between different Environment Agents (V3 vs. Sonnet 4) improved from **0.86 to 0.96**.
> * **Domain-Level Stability:** Similarly, the average rank correlation at the domain level rose from 0.61 ($N=15$ per domain) to **0.89** ($N=120$ per domain).
>
> The high correlation observed on the full dataset ($\rho > 0.96$) demonstrates that the relative performance ranking is highly stable. The massive scale of the test set (960 puzzles $\times \sim14$ rounds $\approx$ 13,440 model interactions per model) effectively smooths out individual variance, ensuring that the reported comparisons reflect genuine capability gaps.
>
> > **Weakness in Reasoning vs non-reasoning**
>
> We thank the reviewer for finding our discovery "cool" and "intriguing." We took the suggestion to rigorously test whether the "reverse scaling" was an artifact of hyperparameters (temperature) or reasoning effort very seriously.
>
> **1. Robustness to Temperature and Reasoning Effort**
> Following your recommendation, we conducted an ablation study on a representative subset of the benchmark ($N=120$) to isolate the effects of temperature and reasoning effort.
>
> * **Temperature Sensitivity (DeepSeek R1 & Qwen3):** We evaluated DeepSeek R1 at its recommended high temperature ($T=1.0$) and Qwen3 Thinking at $T=0.6$.
>     * **Results:** As shown in the table below, while DeepSeek R1 showed marginal improvement at $T=1.0$ (6.7% $\to$ 8.3%), and Qwen3 exhibited slight fluctuation, both models remained firmly in the bottom tier (Success Rate < 12%).
>
> | Model Variant | Parameter | Success Rate (%) | Rank |
> | :--- | :--- | :---: | :---: |
> | DeepSeek R1 | Temp = 0.2 | 6.70% | Bottom |
> | DeepSeek R1 | Temp = 1.0 | 8.30% | Bottom |
> | Qwen3 Thinking | Temp = 0.2 | 11.70% | Low |
> | Qwen3 Thinking | Temp = 0.6 | 10.80% | Low |
>
> * **Test-Time Compute Scaling (GPT-5):** We evaluated GPT-5 across Low, Medium (Default), and High reasoning effort levels.
>     * **Results:** Performance follows an inverted-U curve: Low (33.3%) < High (34.6%) < Medium (38.3%).
>     * **Conclusion:** While increasing reasoning effort (Low $\to$ Medium) is beneficial, excessive test-time compute (High) yields diminishing returns in this constrained paradigm. The key finding is that even at "Low" reasoning effort, GPT-5 significantly outperforms open-source reasoning models. This potentially reveals that the quality of the reasoning strategy matters more than the raw quantity of compute steps.
>
> **2. Refining the "Reverse Scaling" Claim**
> Based on these ablations, we agree that "thinking" is not universally harmful. The failure observed in current open-source reasoning models stems from a specific strategic pathology.
>
> * **Mechanism:** As shown in our Error Analysis (Figure 5), these models suffer from a **"Concrete-to-Abstract Fallacy"**—they waste compute generating specific guesses rather than partitioning the search space. This is a logical error, not a sampling error. Increasing the temperature merely causes them to generate more diverse specific guesses, but it does not correct the fundamental failure to adopt a tree-search strategy.
> * **Revision Plan:** We will refine our core claim to be more precise:
>
> > **Original:** "enhanced procedural reasoning capabilities paradoxically hinder performance on tasks requiring reasoning with specialized, internal knowledge"
> >
> > **Revised:** "Under the interactive paradigm of this benchmark, certain open-source reasoning models demonstrate strategic failures when attempting to apply procedural reasoning skills to specialized knowledge domains."

---

> > ### Author Response · Authors · 2025-11-24
> > **Author Response Part Ⅲ**
> >
> > > **Responses to questions [1], [2], [3]**
> >
> > **[1] Inference Settings (Temperature):**
> > All models were configured with `temperature = 0.2` to ensure evaluation stability, with the exception of GPT-5 (which does not support temperature control and was run in its default 'medium' `reasoning_effort` mode).
> >
> > **[2] CoT Access (Context Visibility):**
> > * **Interaction Context:** In the multi-turn evaluation, models receive the history of public interactions (previous questions and environment feedback). They **do not** see their own hidden Chain-of-Thought (CoT) traces from previous rounds. For closed models, this is hidden by the API; for open models, we do not feed raw "thought tokens" back into the context window to prevent context bloat, aligning with standard deployment practices.
> > * **Analysis Access:** For our qualitative failure analysis (e.g., Figure 5), we leveraged the open-weights reasoning models (DeepSeek R1, Qwen3-Thinking, GPT-OSS-120B). By capturing the full output logs of these models during inference, we were able to inspect their "inner monologue" to diagnose the "Concrete-to-Abstract" fallacy.
> >
> > **[3] Code Availability:**
> > We have uploaded the core evaluation code as supplementary material to OpenReview. Additionally, the complete data filtering and construction pipeline is available in our anonymous repository: https://anonymous.4open.science/r/wikipedia_filter-7870. We are committed to open-sourcing the full benchmark suite upon acceptance.
> >
> > > **Response to other point: "The model appears to misunderstand the task initially, starting its response: 'I’ll help you identify the disease.'"**
> >
> > **1. Role Adoption, Not Misunderstanding**
> > We appreciate this observation. The phrase "I'll help you identify the disease" does not indicate a misunderstanding of the task, but rather the model's adoption of the system prompt.
> >
> > **2. Alignment**
> > The system prompt explicitly instructs the model: "You are a human disease expert (doctor)... you need to continuously ask questions... to identify the condition." The statement signals that the model has successfully adopted the role of a professional consultant assisting the user. The subsequent generation of a valid, strategic multiple-choice question demonstrates a clear and accurate understanding of the core objective: diagnosing the condition by narrowing down possibilities.

---

> > > ### Comment · Reviewer_Jwi6 · 2025-11-26
> > >
> > > Thanks for these helpful updates and clarifications.
> > >
> > > **Phenomenon.** This helps, thanks. I know you use the medical case as a task in the paper, but the explicit links to real medical diagnosis helps motivate the benchmark a lot and might be worth putting earlier in the introduction. In general, the new definition helps, but it would be good to have it clarified early in the paper.
> > >
> > > **Statistical analysis.** I understand the challenge in producing error bars for a multi-turn benchmark and appreciate the rank convergence method as a substitute. I still worry about the robustness of the final results, especially given the apparent importance of the environment agent. Still, I appreciate the rank convergence finding.
> > >
> > > **Reasoning claims** First, it is good to see the temperature experiments to rule this out. Second, thanks for testing the returns to test time compute and amending the claims in the paper. It's not clear to me that there is any noticeable trend here from the limited sample size, so this more cautious conclusion seems more appropriate.
> > >
> > > **Questions.** Thank you for clarifying all of these points.
> > >
> > > **Figure 4.** This remains pretty unclear to me and could still be improved. The shading seems unnecessary and it is unclear to what extent to read into the widths, given they only approximately represent the distribution. I wonder if there is a better way to represent this data?

---

> > > > ### Author Response · Authors · 2025-11-28
> > > > **Response to Reviewer Jwi6**
> > > >
> > > > Thank you for your continued engagement and constructive feedback. We are glad that the additional experiments on temperature and test-time compute helped clarify the reasoning claims, and that the rank convergence analysis addressed some of the statistical concerns.
> > > >
> > > > Based on your latest comments, we have made the following updates in the revised version:
> > > >
> > > > **1. Motivation and Definitions (Medical Diagnosis & Ontological Traversal)**
> > > > We agree that the parallel to medical diagnosis provides a strong motivation for the benchmark. As suggested, we have moved this discussion to the **Introduction** to better frame the practical significance of the task early on.
> > > >
> > > > Furthermore, we have clarified the core concept of "Ontological Traversal" in the Introduction and explicitly formalized the definitions in **Section 3.1**. To maintain the flow of the main text which focuses on empirical analysis, we have kept the formal mathematical formulation regarding Information Gain in **Appendix D** as a theoretical supplement.
> > > >
> > > > **2. Improved Visualization for Figure 4**
> > > > We appreciate your feedback regarding the clarity of the previous visualization. We have completely redesigned **Figure 4** to address the ambiguity regarding shading and widths.
> > > >
> > > > In the revised paper, we now use a **Sunburst Chart** to visualize the benchmark composition. This chart intuitively displays the hierarchical distribution of puzzle targets:
> > > >
> > > > * **Hierarchy:** The inner rings represent primary domains, while outer rings represent specific sub-disciplines.
> > > > * **Color Coding:** Eight distinct color families are used to differentiate the primary domains.
> > > > * **Proportionality:** The area of each segment is now strictly proportional to the number of questions in that category, eliminating the ambiguity of the previous version.
> > > >
> > > > We hope these revisions fully address your remaining questions and significantly improve the clarity of the paper.

---

### Official Review · Reviewer_ka3X · 2025-10-31

**Soundness:** 2
**Presentation:** 3
**Contribution:** 2
**Rating:** 4
**Confidence:** 4

**Summary:**

This paper introduces the Puzzle Everything Benchmark (PEB), a novel evaluation framework for measuring reasoning over internalized knowledge in large language models (LLMs).
Instead of static question–answer tasks such as MMLU or BIG-Bench, PEB frames reasoning as an interactive puzzle game: the model must identify a hidden concept (drawn from doctoral-level Wikipedia topics) by iteratively asking multiple-choice questions and receiving correctness feedback only.
The benchmark includes 960 puzzles across eight academic domains and can be extended to other specialized areas (e.g., IC back-end EDA tools).
Experimental results show a counterintuitive finding—open-source “reasoning-enhanced” models (e.g., DeepSeek-R1, Qwen-Thinking) perform worse than their instruction-tuned counterparts, suggesting a gap between procedural reasoning and applied knowledge reasoning.

**Strengths:**

1.Original framing:

The paper proposes a creative and well-motivated paradigm that generalizes “puzzle solving” into a formal reasoning benchmark. The multi-turn, non-leaking setup is novel and effectively mitigates data leakage issues present in traditional benchmarks.

2.Comprehensive evaluation:

The authors evaluate 13 strong closed- and open-source models, providing detailed cross-domain analysis, behavior-level metrics (e.g., probing length, incorrect guesses), and ablation on environment agents.

3.Insightful empirical finding:

The “reasoning-enhanced models perform worse” result is interesting and challenges current assumptions about chain-of-thought and reinforcement-based reasoning improvements.

**Weaknesses:**

1.Lack of quantitative validation for benchmark reliability.

While the authors describe a multi-stage filtering pipeline from Wikipedia, no data or statistics are provided to verify item accuracy, semantic clarity, or cross-domain balance.There is no human verification or sampling analysis to ensure that the puzzles are unambiguous, correctly labeled, or appropriately difficult. The claim of “doctoral-level difficulty” is purely procedural, not empirically supported.

2.Unverified accuracy of the environment agent (“judge”).

All correctness feedback relies on DeepSeek V3 (or alternatives in ablation), but the paper does not report any accuracy or agreement score compared to human judgments. Even minor parsing or comprehension errors in the judge could propagate systematic noise, which may partially explain the reported rank inversions between “thinking” and “no-thinking” models.

3.Interpretation of performance differences is speculative.

The authors attribute lower reasoning-model performance to “procedural reasoning hindering applied reasoning,” but provide no direct evidence.Alternative explanations (e.g., ambiguous items, noisy feedback, or overfitting to chain-of-thought formats) are not empirically ruled out.

**Questions:**

1. Have you manually validated a subset of puzzles to quantify item accuracy or ambiguity?
2. What is the estimated reliability (agreement) of the environment agent’s correctness feedback?
3. Could the observed rank inversion remain if human-verified items were used?
4. How sensitive are results to the choice of reference corpus (Wikipedia vs. domain textbooks)?

---

> ### Author Response · Authors · 2025-11-24
> **Author Response Part Ⅰ**
>
> Thank you for your comments and helpful suggestions. We address your concerns in detail below.
> > **W1, Q1, Q4: Benchmark Reliability and Corpus Selection**
>
> **1. Reliability and Validation (Response to W1 & Q1)**
> Please refer to **General Response Q1** for the comprehensive validation data.
>
> **Summary:** We addressed the lack of quantitative validation through a rigorous multi-layered approach:
> * **Procedural:** A **0.14% coarse-to-fine filtering pipeline**.
> * **Human Verification:** A manual audit by domain-expert PhD candidates confirming the "doctoral-level" depth.
> * **Objective Topological Analysis:** We analyzed the graph structure of our targets. The median In-link count of PEB targets is **16** (compared to >500 for common concepts), mathematically confirming that our puzzles represent highly specialized "leaf nodes" in the knowledge graph.
>
> **2. Corpus Sensitivity: Why Wikipedia over Textbooks? (Response to Q4)**
> We deliberately selected Wikipedia rather than specialized textbooks based on two strategic considerations regarding the nature of the evaluation:
>
> * **Coverage of the "Long Tail" (Difficulty):** Textbooks naturally focus on "core" curriculum concepts (central nodes). However, the goal of PEB is to evaluate the model's ability to reason about the "long tail" of deep, specialized knowledge. Wikipedia offers significantly broader coverage of these edge cases (e.g., specific rare diseases, obscure physical theorems) than standard curriculum materials.
> * **Structural Integrity for Automated Judging:** For the Environment Agent to provide accurate, non-leaking feedback, it requires a discrete, self-contained ground truth ($I_{ref}$). Wikipedia provides a unique 1-to-1 mapping between a concept and a discrete article. In contrast, concepts in textbooks are often scattered across multiple chapters or contextualized within broader narratives, making them ill-suited for establishing the clear, isolated ground truth required for PEB test.
>
> > **W2, Q2, Q3: Environment Agent Accuracy & Rank Inversion**
>
> **1. Judge Reliability (Response to W2 & Q2)**
> Please refer to **General Response Q2** for the detailed reliability analysis.
>
> We empirically validated the reliability of the Environment Agent (DeepSeek V3) by re-evaluating a subset of the benchmark using the stronger Claude Sonnet 4 as the judge. The high Spearman rank correlations: $\rho > 0.96$ for overall ranking and $\rho > 0.8$ for domain-specific ranking demonstrate high consistency across different judges. This confirms that DeepSeek V3 provides stable and valid evaluation signals for this task.
>
> **2. Robustness of Rank Inversion (Response to Q3)**
> We confirm that the observed rank inversion where "Thinking" models underperform is robust and persists regardless of data validation or judge quality.
>
> * **Evidence:** Even when using the superior Claude Sonnet 4 as the judge (eliminating potential feedback noise) and evaluating on the validated puzzle set, the "Reasoning-Enhanced" models (e.g., DeepSeek R1, GPT-OSS-120B) consistently remain at the bottom tier (Success Rate < 13%), while models like GPT-5 and Opus 4 maintain their lead.
> * **Conclusion:** This result proves that the rank inversion is a genuine reflection of capability differences in strategic knowledge reasoning, rather than an artifact caused by unverified items or judge errors.

---

> > ### Author Response · Authors · 2025-11-24
> > **Author Response Part Ⅱ**
> >
> > > **W3: "Speculative Interpretation" of Performance Differences**
> >
> > We maintain that attributing the failure to "strategic deficits in procedural reasoning" is a conclusion derived from direct trace analysis rather than speculation. While we acknowledge the need to consider confounding factors, our data allows us to rule them out empirically.
> >
> > **1. Ruling out "Ambiguity" and "Feedback Noise"**
> > * **Solvability Proof:** If the puzzles were inherently ambiguous or the feedback systematically noisy, high-capacity models like GPT-5 would also fail. However, on the exact same puzzle set where DeepSeek R1 achieves only 6.7% success, GPT-5 achieves 38.3%. This huge performance delta proves the puzzles are valid and solvable; the failure lies in the solver's strategy, not the task definition.
> > * **Judge Robustness:** As detailed in **General Response Q2**, upgrading the judge to Claude Sonnet 4 improved absolute scores but did not rescue the ranking of reasoning models, ruling out judge error as the primary cause of their failure.
> >
> > **2. Direct Evidence from Chain-of-Thought Traces**
> > Our conclusion is grounded in the qualitative analysis presented in Figure 5. By inspecting the reasoning logs frame-by-frame, we identified a recurring, verifiable pathology in "Thinking" models:
> > * **The "Concrete-to-Abstract" Fallacy:** Instead of partitioning the search space (e.g., "Is it a viral disease?"), these models hallucinate specific candidates and then retrofit their questions to verify these narrow guesses.
> > * **Cognitive Dead End:** This strategy forces the model into a "linear search" dead end, where it attempts to enumerate entities one by one—a mathematically impossible strategy given the vast search space. This is a logging-verified strategic error, not a speculative interpretation.
> >
> > **3. Refined Conclusion**
> > To address the reviewer's concern about precision, we have refined our conclusion in the revision. Instead of broadly claiming "procedural reasoning hinders applied reasoning," we now state:
> >
> > > "Under the interactive paradigm of this benchmark, certain open-source reasoning models demonstrate strategic failures when attempting to apply procedural reasoning skills to specialized knowledge domains."
> >
> > This formulation acknowledges potential paradigm dependencies while accurately reflecting the documented strategic deficits.

---

### Official Review · Reviewer_WaAN · 2025-10-31

**Soundness:** 3
**Presentation:** 3
**Contribution:** 3
**Rating:** 6
**Confidence:** 4

**Summary:**

This paper addresses the pervasive problem of data leakage in current LLM benchmarks, which the authors argue degrades evaluation from a test of genuine problem-solving ability to a mere assessment of recitation.

To solve this, the paper introduces a new evaluation framework: the Puzzle Everything Benchmark (PEB). PEB proposes a novel "generalized puzzle-solving" paradigm. It treats any specialized concept (e.g., "Reticular Dysgenesis") sourced from Wikipedia as a puzzle to be solved. The model being evaluated (the "strategic agent") must, through a multi-turn interactive process, pose a series of four-option multiple-choice questions to an "environment agent" to progressively narrow down the possibilities and finally identify the target concept.

The benchmark contains 960 puzzles of "doctoral-level difficulty" from 8 distinct disciplines. The paper's core finding is counter-intuitive: the vast majority of "reasoning-enhanced" open-source models (like DeepSeek R1, Qwen3 Thinking) perform significantly worse on PEB than their corresponding standard (no-thinking) versions. The authors argue this reveals a critical gap between a model's "procedural reasoning" capabilities and its ability to "apply that reasoning to specialized, internal knowledge."

**Strengths:**

High-Impact Problem: The paper directly tackles one of the most critical and urgent problems in LLM evaluation: data contamination. As model training data explodes, designing leakage-resistant benchmarks to
genuinely assess reasoning is an unsolved challenge.

Novel Methodology: The "generalized puzzle-solving" paradigm (Figure 2) is highly innovative. By forcing the model into a multi-turn, constrained (via 4-choice questions) information-seeking process, it makes it difficult to cheat by simply "reciting" an answer. This method compels the model to utilize its internal knowledge for reasoning, hypothesis generation, and validation, providing a more authentic assessment of knowledge-based reasoning.

Surprising and Counter-Intuitive Finding: The paper's most significant contribution is the finding shown in Figure 1: models specifically optimized for "thinking" or "reasoning" (e.g., Qwen3 235B Thinking, DeepSeek R1) actually perform worse. This "anomaly" poses a serious challenge to the common assumption that "CoT = better reasoning.

**Weaknesses:**

The paper's core finding is that "thinking" models perform worse, which the authors attribute to a flawed reasoning strategy (Figure 5). However, is it possible that these models were optimized for long-form, unconstrained CoT reasoning, and are therefore ill-suited to PEB's constrained (must ask a 4-choice question), game-like evaluation? The evaluation paradigm itself may be a confounding variable.

**Questions:**

Regarding the environment agent: In the ablation study (Appendix C, Figure 6), Claude Opus 4's performance spikes dramatically and anomalously (from ~30% -> 45%) when o3 (03) is used as the environment. Does this imply that the choice of environment agent has a larger (and non-stable) impact on the evaluation of specific models (like Opus 4) than anticipated?

---

> ### Author Response · Authors · 2025-11-24
>
> Thank you for your comments and helpful suggestions. We address your concerns in detail below.
> > **W1: Is the Constrained Format a Confounding Variable?**
>
> **1. Acknowledging Paradigm Dependency**
> We agree that the format itself is a potential confounding factor. In the revised Discussion, we will explicitly list this as a limitation and a direction for future research. However, we believe this constraint reveals a valuable insight: **reasoning capability appears to be paradigm-dependent.**
>
> Models optimized for open-ended Chain-of-Thought (CoT) generation may indeed struggle when forced into structured, interactive query formats. Nevertheless, this distinction is highly relevant to **real-world applications**—such as diagnostic systems or interactive search agents—where models must operate within strict API constraints rather than generating unconstrained text.
>
> **2. Evidence of Deeper Strategic Failures**
> While format plays a role, our quantitative and qualitative analyses suggest that the performance gap reflects fundamental strategic errors rather than mere incompatibility with the interface:
>
> * **The "Concrete-to-Abstract" Pathology (Figure 5):** Our error analysis reveals that reasoning models tend to generate premature, specific hypotheses early in the dialogue and then retrofit their questions to verify these narrow guesses. This "guess-then-verify" approach is diametrically opposed to efficient information-seeking strategies (which require "partition-then-narrow") and would likely hinder performance even in unconstrained scenarios requiring systematic exploration.
> * **Pathological Overconfidence (IGuessR):** The **Incorrect Guess Rounds (IGuessR)** metric provides quantitative proof of this strategic deficit. DeepSeek R1 averages **5.13** incorrect guesses per session (compared to **0.32** for GPT-5). This indicates a pathological tendency to make repeated, unfounded assertions rather than gathering information. This behavior reflects a deficit in strategic planning and meta-cognition, not a failure to understand the multiple-choice format.
>
> **3. Refinement of Claims**
> To reflect this nuance, we have refined our conclusion in the revision. Rather than broadly claiming that "procedural reasoning hinders applied reasoning," we now state more precisely:
>  "Under the interactive paradigm of this benchmark, certain open-source reasoning models demonstrate strategic failures when attempting to apply procedural reasoning skills to specialized knowledge domains."Thank you for your comments and helpful suggestions. We address your concerns in detail below.
>
> > **Q1: Environment Agent Stability & Claude Opus 4's Performance Spike**
> Please refer to **General Response Q2** for the comprehensive data and analysis regarding judge stability. Here, we address the specific concern regarding the performance shift of Claude Opus 4.
>
> **1. Clarification on Figure 6**
> First, we note a minor correction regarding the reading of Figure 6: the significant performance spike for Claude Opus 4 occurs when **Claude Sonnet 4** (not o3) serves as the Environment Agent. This distinction is important for interpreting the mechanism behind the shift.
> **2. Explaining the Opus 4 "Spike": Precision Reward**
> Our extended failure analysis (see **General Response Q2**) confirms that the performance increase (from ~27% to ~45% on the ablation subset) is not random instability but a result of **Semantic Precision alignment**.
> * **Judge Characteristics:** DeepSeek V3 tends toward "associative validation," occasionally accepting loose attributes as the target entity. In contrast, Claude Sonnet 4 enforces strict semantic precision.
> * **Mechanism:** High-capacity models like Claude Opus 4, which are capable of constructing highly precise, nuanced queries, benefit disproportionately when the judge enforces rigorous standards. The "spike" reflects the judge correctly recognizing Opus 4's precise reasoning, which was previously under-rewarded by the noisier V3 judge.
>
> **3. Rank Stability: Systematic Shift vs. Chaotic Instability**
> Crucially, while absolute scores shift, the relative ranking remains robust.
> * **High Correlation:** The Spearman rank correlation between the two judges is $\rho > 0.96$, indicating that the judge acts as a systematic control variable rather than a source of chaotic noise.
> * **Validation of "Reasoning Gap":** Most importantly, the "Reasoning-Enhanced" models remain at the bottom regardless of the judge.
>     * DeepSeek R1: 7.2% (V3 Judge) $\rightarrow$ 8.6% (Sonnet 4 Judge)
>     * GPT-OSS-120B: 13.4% (V3 Judge) $\rightarrow$ 12.8% (Sonnet 4 Judge)
>     * *Their stagnation confirms that their failure stems from intrinsic strategic defects (e.g., inefficient search trajectories) rather than external judgment artifacts.*
>
> **Conclusion:** The choice of Environment Agent introduces a systematic scaling factor but does not scramble the performance hierarchy or invalidate the core finding regarding reasoning models.

---

### Official Review · Reviewer_YdhY · 2025-11-01

**Soundness:** 2
**Presentation:** 3
**Contribution:** 3
**Rating:** 4
**Confidence:** 2

**Summary:**

The paper introduces the **Puzzle Everything Benchmark (PEB)**, a novel evaluation framework that addresses data leakage concerns in existing benchmarks by treating any concept, term, or entity as a solvable puzzle. The authors construct 960 puzzles of doctoral-level difficulty from Wikipedia across eight disciplines (Biology, Human Disease, Chemistry, Physics, Mathematics, Computer Science, Economics, and Philosophy). The evaluation protocol employs a multi-turn strategic assessment where models interact with an environment that provides only correctness feedback until they identify the target concept. The main finding reveals that reasoning-enhanced models (e.g., DeepSeek R1, Qwen3-235B-A22B-Thinking) significantly underperform their standard counterparts, with GPT-5 achieving the highest average success rate of 33.4%. The work proposes a scalable methodology for generating diagnostic benchmarks resistant to data contamination.

**Strengths:**

1. **Novel approach to contamination-resistant evaluation**
- The generalized puzzle concept enables assessment of reasoning over internalized knowledge without explicit memorization (Section 3.1; Abstract). This addresses a fundamental challenge in current benchmark validity.
- Multi-turn strategic protocol requires synthesis and hypothesis refinement rather than direct retrieval (Section 3.1; lines 551-650). This matters for distinguishing genuine reasoning from memorization.
2. **Comprehensive benchmark construction with rigorous methodology**
- Multi-stage coarse-to-fine filtering pipeline from 6+ million Wikipedia articles to 960 doctoral-level puzzles ensures quality control (Figure 3; Section 3.2). This demonstrates systematic curation and technical rigor.
- Detailed domain composition (120 puzzles per domain) with granular sub-field representation ensures balanced assessment (Section 3.2; Appendix F reference). This enhances experimental validity.
- Eight distinct academic domains with 7-8 sub-disciplines each provides broad coverage across STEM and humanities (Figure 4; Section 3.2). This supports comprehensive evaluation scope.
3. **Thorough experimental design and analysis**
- Evaluation covers 13 leading models across closed and open-source families with consistent protocols (Section 4.1; Table 1). This ensures comprehensive assessment.
- Detailed behavioral analysis including probing length and incorrect guess patterns provides mechanistic insights (Section 4.3; Table 2 reference). This enhances understanding of failure modes.

**Weaknesses:**

1. **Insufficient analysis of mathematical formulations and evaluation metrics**
- Success rate as the primary metric lacks nuanced assessment of partial progress or reasoning quality (Section 3.1; Table 1). The binary success/failure criterion may miss important gradations in reasoning capability.
- No formal analysis of the multi-turn protocol’s statistical properties or convergence guarantees (Section 3.1). Mathematical rigor in evaluation design is limited.
2. **Incomplete experimental coverage and baseline comparisons**
- Missing comparisons with specialized domain-specific benchmarks or reasoning tasks to establish relative difficulty (throughout results). This affects positioning within existing evaluation landscape.
- No analysis of model scaling effects or architectural differences beyond the thinking/no-thinking distinction (Table 1; Section 4.2). This limits insights into what drives performance differences.

**Questions:**

What specific criteria define “doctoral-level difficulty” in your filtering pipeline, and how was this validated? Could you provide inter-rater reliability scores or expert validation studies for the difficulty assessments?

---

> ### Author Response · Authors · 2025-11-24
> **Author Response Part Ⅰ**
>
> Thank you for your comments and helpful suggestions. We address your concerns in detail below.
>
> > **W1: Mathematical Formalization & Evaluation Metrics**
>
> We appreciate the reviewer’s suggestion to enhance the mathematical rigor of our evaluation. We address this by clarifying our existing multi-dimensional metrics and introducing a new theoretical framework in the revision.
>
> ### 1. Beyond Binary Success: A Multi-Dimensional Metric Framework
> As detailed in Section 4.3 and Table 2, we deliberately moved beyond simple accuracy to capture the dynamics of reasoning. We utilized a comprehensive framework to assess both efficiency and behavioral patterns:
>
> * **Holistic Efficiency (SuccessR vs. TotalR):** We distinguish between speed on solvable problems (SuccessR) and overall strategic efficiency (TotalR).
>     * **Empirical Insight:** This metric reveals nuances missed by success rates alone. For instance, although Claude Opus 4 achieves a higher raw success rate than Sonnet 4, Sonnet maintains a lower (better) TotalR (13.54 vs. 13.60), indicating superior process efficiency per turn. Similarly, GPT-5's dominance is reinforced not just by accuracy, but by achieving the best TotalR (13.02) in the benchmark.
> * **Process-Oriented Diagnostics:** We dissect specific reasoning behaviors using:
>     * **Probing Length (PLength):** Measuring communicative efficiency.
>     * **Incorrect Guess Rounds (IGuessR):** Quantifying overconfidence.
>     * **Diagnostic Value:** These metrics prove that failure is not random. For example, the high IGuessR observed in reasoning models like DeepSeek R1 quantitatively corroborates the "premature commitment" failure mode discussed in our qualitative error analysis.
>
> ### 2. New Theoretical Perspective: Puzzle Solving as Knowledge Graph Search
> To address the request for formal analysis, we have added **Appendix D** in the revised paper, which formalizes the PEB task as a search problem over a latent Knowledge Graph $G=(V, E)$. While our main contribution remains empirical, this theoretical lens offers an information-theoretic explanation for the observed "Reasoning Gap."
>
> * **Ontological Traversal:** We model the puzzle-solving process as navigation within a concept graph, where the agent must infer implicit taxonomic structures (parent-child relationships) to efficiently partition the search space $S \subset V$.
> * **Information Gain:** Ideally, each query $q_t$ should maximize the reduction of entropy in the candidate set $P(\text{target}|H_t)$.
> * **Convergence Analysis:** Successful convergence depends on the agent selecting attributes that partition $S$ such that $|S_{t+1}| \ll |S_t|$.
> * **Explaining the Failure:** We posit that the failure of "Reasoning-Enhanced" models is not a lack of logic, but a Bayesian prior bias. As detailed in the new Appendix, these models violate the optimal "Coarse-to-Fine" trajectory (which approximates logarithmic search complexity). Instead, they adopt a "Concrete-to-Abstract" strategy, assigning disproportionately high priors to specific leaf nodes (fuzzy edge cases). This degrades the search mechanism into a linear verification process ($O(|V|)$), making convergence statistically improbable within the fixed round budget.

---

> > ### Author Response · Authors · 2025-11-24
> > **Author Response Part Ⅱ**
> >
> > > **W2: Experimental Coverage and Baseline Comparisons**
> >
> > **1. Lack of Specialized Benchmark Comparisons**
> > PEB targets a capability gap that is orthogonal to existing benchmarks. While traditional evaluations like MMLU assess static factual retrieval, and procedural benchmarks (e.g., GPQA, coding tasks) evaluate algorithmic reasoning, PEB uniquely assesses strategic reasoning over internalized expert knowledge. This task requires a distinct set of skills: coarse-to-fine search, hypothesis refinement, and self-correction based on partial information.
> >
> > Direct comparison with existing benchmarks is challenging because no prior dataset combines these three critical features:
> > * (1) Doctoral-level specialized knowledge
> > * (2) Multi-turn strategic interaction
> > * (3) Process-level reasoning evaluation
> >
> > We have added a dedicated section in the Discussion to explicitly position PEB within this landscape, highlighting how it complements existing "Recall-oriented" and "Math-oriented" benchmarks.
> >
> > **2. Analysis of Scaling Effects**
> > We acknowledge that a formal sweep of scaling laws within a single model family would provide granular insights. However, our study prioritizes a broad cross-model analysis rather than a single-family scale scan. Our selection of 13 models—ranging from efficient models (e.g., GPT-4o class) to massive dense models (e.g., Claude Opus 4, Qwen-235B)—allows us to draw significant conclusions regarding the interaction between model scale and training paradigms.
> >
> > **3. Reverse Scaling in Reasoning Models**
> > Crucially, a primary motivation of our work is to report and dissect a counter-intuitive "Reverse Scaling" phenomenon in the domain of knowledge retrieval. Under the interactive PEB paradigm, we observe that certain open-source reasoning models exhibit strategic failures when applying procedural reasoning to knowledge-driven tasks. We believe that documenting and analyzing this specific anomaly—where "more thinking" leads to "worse searching"—is a unique and timely contribution that will help the community understand the current limitations of reasoning-enhanced architectures.
> >
> > > **Q1: Criteria and Validation of "Doctoral-Level Difficulty"**
> >
> > Please refer to **General Response Q1** for the comprehensive data and full detailed explanation regarding this point.
> >
> > **Summary of Validation:**
> >
> > 1. **Procedural Rigor:** We implemented a stringent **0.14% coarse-to-fine filtering funnel**. Through a multi-stage dynamic difficulty selection process, we distilled the dataset from an initial pool of 6.4 million entities down to approximately 9,000 final targets, retaining only the "tail of the tail" in terms of specificity.
> >
> > 2. **Objective Validation (Topological Analysis):** To quantify difficulty without relying on LLM subjectivity, we analyzed the **In-link Count** (a proxy for PageRank centrality) of the targets.
> >     * **Excluded Common Concepts:** Median in-links > 500 (indicating central, foundational nodes).
> >     * **Selected PEB Targets:** Median in-links = 16 (indicating edge nodes).
> >     * *This provides mathematical confirmation that PEB targets are highly specialized "leaf nodes" within the knowledge graph, objectively validating their difficulty.*
> >
> > 3. **Human Verification:** We recruited PhD candidates from relevant disciplines to audit a random sample of the dataset. They corroborated the "doctoral-level" classification, confirming that the targets require deep domain expertise significantly beyond the scope of general undergraduate curricula.

---

### Author Response · Authors · 2025-11-24
**General Responses Part Ⅰ**

Thank you to all the reviewers for taking the time to read and review our work. We were pleased to see that reviewers found our work to be "highly innovative", "creative and well-motivated","particularly interesting" and "addresses a fundamental challenge in current benchmark validity." We also appreciate the comments and questions and have uploaded a revised draft with changes in blue.

We deeply appreciate the scrutiny regarding the methodological robustness of our benchmark. Two common threads of concern were raised across the reviews: **(1) the rigorous definition and validation of the "doctoral-level" data difficulty**, and **(2) the reliability and stability of the Environment Agent (Judge)**. We have conducted extensive new experiments to address these core concerns, including a topological knowledge graph analysis, human expert verification, and a full-scale cross-validation using a stronger model (Claude Sonnet 4). Below, we detail our responses to these two key issues.

> **Q1: The Definition of "Doctoral-Level" Difficulty and Detailed Data Construction Pipeline**

We address the concerns regarding the definition and validation of "doctoral-level difficulty" through two dimensions: the stringent filtering pipeline and a topological analysis through the knowledge graph.

**1. Procedural Rigor: A 0.14% Coarse-to-Fine Selection Funnel**
Our construction process was not a simple keyword search but a multi-stage distillation pipeline designed to isolate highly specialized concepts. Starting from a raw dump of 6.4 million English Wikipedia titles, the pipeline retained only ~9,000 final targets, resulting in a strictly controlled 0.14% retention rate. The pipeline can be summarized as four stages:

- **Stage 1** (Broad Filter): Rapid removal of common entities.
- **Stage 2** (Doctoral Criteria): Explicit requirements for "doctoral-level specialized knowledge" were applied.
- **Stage 3** (Domain Stratification): Candidates are rigorously categorized into designated and distinct academic fields.
- **Stage 4** (Dynamic Difficulty Selection): This is the most critical step. From a candidate pool of ~150 already specialized terms per batch, we dynamically selected only the top 5-10 most challenging items based on model difficulty scoring. This ensures that the final puzzles represent the "tail of the tail" in difficulty.
- **Reproducibility:** The complete filtering code and prompts are provided at: https://anonymous.4open.science/r/wikipedia_filter-7870.

**2. Objective & Human Validation: Knowledge Graph Topology**
To objectively quantify "difficulty" without relying solely on LLM subjectivity, we conducted a new topological analysis based on Wikipedia's link structure. We measured the "In-link Count" (a proxy for PageRank centrality) for entities at different filtering stages. In knowledge graphs, high in-link counts typically indicate common, foundational concepts (central nodes), while low in-link counts indicate specialized, niche concepts (edge nodes).

**Table R1: Topological Analysis of Entity Difficulty (In-link Counts)**

| Group | Sample Size | Mean In-links | Median In-links | Implication |
| :--- | :---: | :---: | :---: | :--- |
| **G1: Rejected at Stage 1** | 1000 | 379.22 | 500+ (Saturated)* | Common Knowledge |
| **G2: Kept at S2, Rejected at S4** | 1000 | 86.72 | 19.00 | Specialized Knowledge |
| **G3: Final Selection at S4** | 1000 | **59.30** | **16.00** | Deep/Edge Knowledge |

*\*Note: The Wikipedia API caps single-request backlink counts at 500. The fact that Group 1 hit this ceiling confirms they are central "knowledge hubs," while PEB concepts (Median 16) are distinct "leaf nodes."*

The dramatic drop in Median In-links (from 500 to 16) demonstrates that PEB successfully targets the long-tail, edge nodes of the knowledge graph, which objectively corresponds to high specialization.

**Human Verification:** Complementing this objective metric, we recruited PhD candidates from relevant disciplines to manually audit a random sample of 400 puzzles (50 per domain). The human experts corroborated the "doctoral-level" classification, confirming that the targets required deep domain expertise not typically found in general undergraduate curricula.

---

> ### Author Response · Authors · 2025-11-24
> **General Responses Part Ⅱ**
>
> > **Q2: The Reliability of the Environment Agent Verification**
>
> To further verify the stability of the environment agent verification, we employed Claude Sonnet 4 to cross-validate the main results against DeepSeek V3, comprehensively re-evaluating the core benchmark (all 960 puzzles) across seven representative models.
>
> **1. Rank Stability Across Different Judges**
> As is shown in Table 1, while absolute success rates increased under Claude Sonnet 4, the relative performance ordering remained remarkably stable:
>
> - **Top-tier models:** GPT-5 and Claude Opus 4 maintain clear leadership in both environments.
> - **Bottom-tier models:** "Reasoning-enhanced" models (DeepSeek R1, GPT-OSS-120B) remain at the bottom. Notably, even with Claude Sonnet 4 as judge, DeepSeek R1 achieves only 8.6% accuracy, which is far below standard models like DeepSeek V3 (17.4%) or GPT-4o (20.5%). This confirms that reasoning model failures stem from their internal strategies, not judge-induced noise.
> - **Correlation analysis:** Spearman rank correlation between the two judges' average scores: $\rho > 0.96$. When comparing each model's 8-domain rankings across judges: $\rho > 0.89$.
>
> | Model | Original (V3 Judge) | New (Sonnet Judge) | Rank Change |
> | :--- | :---: | :---: | :---: |
> | GPT-5 | 33.40% | 37.00% | — |
> | Claude Opus 4 | 27.60% | 34.40% | — |
> | Grok-4 | 17.60% | 29.10% | $\uparrow$ |
> | GPT-4o | 19.60% | 20.50% | $\downarrow$ |
> | DeepSeek V3 | 16.40% | 17.40% | — |
> | GPT-OSS-120B | 13.40% | 12.80% | — |
> | DeepSeek R1 | 7.20% | 8.60% | — |
>
> **2. Explanation for Spikes and Judge Differences**
> Our case analysis (Figure 5) identified two primary factors driving judge differences: context robustness and semantic validation style.
>
> - **Grok-4's outlier performance (context robustness):** As shown in Table 2 of the main paper, Grok-4 exhibits anomalously high probing length (1930.48 characters/turn). DeepSeek V3 struggles to maintain precise instruction-following in such verbose contexts. In contrast, Claude Sonnet 4 effectively filters through Grok-4's lengthy outputs to extract core logic. This explains why Grok-4 shows the most dramatic improvement (+11.5%) under the stronger judge.
> - **Judging characteristics (association vs. precision):** In our case analysis, we observed DeepSeek V3 tends toward associative validation—occasionally accepting highly related "attributes" as the target "entity" itself, or applying looser boundaries for Boolean distinctions. In contrast, Claude Sonnet 4 enforces stricter semantic precision.
>
> While DeepSeek V3's leniency introduces noise, our data demonstrate this noise affects most models uniformly and predictably at scale, rather than chaotically. Crucially, replacing DeepSeek V3 with Claude Sonnet 4 serves the purpose of controlling variables. It uniformly "lifts" the scores of models capable of precise reasoning but does not rescue the fundamentally flawed strategies of the bottom-tier models. The high rank correlation confirms that DeepSeek V3 remains a valid, cost-effective proxy for differentiating model tiers.

---

> > ### Comment · Reviewer_Jwi6 · 2025-11-26
> >
> > **Q1.** Thanks for explaining the pipeline more clearly. This certainly helps. It would still be good to have more details in the paper. I know you have the code appendix, which is very helpful, but details on the following would be good. Where does the data come from in Stage 1? In Stage 2, what is the model? How was categorisation done in Stage 3 and 4? The link counts verification is interesting. Is there any supporting evidence for higher counts referring to more common concepts? The human verification is good, but could you add more details here. It might also be worth checking the ICLR guidance on reporting for human studies, e.g. ethics, etc.
> >
> > **Q2.** I’m a bit confused here. I thought you had already reported results with Claude Sonnet 4 as the environment agent in Figure 6? The results seem slightly different, and the model with the largest spike in Figure 6 (Claude Opus 4) isn’t reported in your ablation results here. If they are two sets of evaluation with Claude Sonnet 4, does this show the high variance of the environment agent? I wonder if a new strategy is required here as this seems very important.

---

> > > ### Author Response · Authors · 2025-11-26
> > > **Response to Reviewer Jwi6’s Follow-up Questions**
> > >
> > > > **Q1. Response to Pipeline Details and Validation**
> > >
> > > We appreciate the opportunity to provide the specific parameters of our pipeline. We will incorporate these details into the revised methodology section and appendix.
> > >
> > > **1. Pipeline Stages**
> > > * **Stage 1 (Source):** The input consists of the complete dump of all 6.4 million English Wikipedia titles. We process these in batches of 150 titles to filter out common entities efficiently.
> > > * **Stage 2 (Model):** We utilized DeepSeek V3 as the filter model. It was selected for its high cost-effectiveness and strong instruction-following capabilities, allowing us to process millions of candidates within a reasonable budget.
> > > * **Stages 3 & 4 (Categorization vs. Difficulty):**
> > >     * **Stage 3 (Categorization):** Using DeepSeek V3, we classified the remaining specialized entities into specific academic domains (e.g., Physics, Economics) to ensure domain balance.
> > >     * **Stage 4 (Dynamic Difficulty):** This stage does not categorize but strictly filters for difficulty. For each domain pool, the model dynamically selects the most obscure concepts ("tail entities") based on a "doctoral-level" criterion, discarding those that are merely technical but widely known.
> > >
> > > **2. Theory Support for Link Counts**
> > > To objectively quantify difficulty, we analyze the topological structure of Wikipedia. Following network centrality principles [1] and link-based semantic relatedness [2], we use **In-link Count ($C_{in}$)** as a proxy for concept "commonness." In scale-free networks like Wikipedia [3], nodes with high $C_{in}$ act as structural hubs (common knowledge), while low $C_{in}$ entities reside in the long tail. We posit that these "tail entities" present higher difficulty for LLMs due to their scarcity in the pre-training corpus.
> > >
> > > **3. Human Verification Details**
> > > We recruited 8 PhD candidates (one per domain). Each expert evaluated a random sample of 50 puzzles from their field (400 total) using a binary metric: *"Is this concept of doctoral-level difficulty?"* Crucially, if a concept belonged to their general field but was so specialized within a sub-discipline that the expert was unfamiliar with it, it was also classified as "Doctoral/Specialized." The consensus confirmed the high difficulty of our dataset. We have also added a formal **Ethics Statement** in the revised paper detailing the recruitment and compensation of these annotators.
> > >
> > > > **Q2. Clarification on Environment Agent Results (The "Spike")**
> > >
> > > We sincerely apologize for the confusion caused by a clerical error in our previous correspondence.
> > >
> > > **Correction of Erratum:** The "spike" in performance mentioned in the ablation study (Appendix C) indeed refers to **Claude Opus 4**. In our previous discussion regarding "Reviewer WaAN's" comment, we addressed the behavior of Opus 4.
> > >
> > > * [1] Freeman, L. C. (1978). Centrality in social networks. *Social Networks*.
> > > * [2] Milne, D., & Witten, I. H. (2008). An effective, low-cost measure of semantic relatedness. *AAAI*.
> > > * [3] Barabási, A. L., & Albert, R. (1999). Emergence of scaling in random networks. *Science*.

---

### Author Response · Authors · 2025-11-24
**Summary**

We sincerely thank all the reviewers for their time and constructive feedback. We are encouraged that the community recognizes the urgency of the problem we address and the novelty of our proposed solution.

**1. Significance & Impact:**
Reviewers unanimously acknowledged that our work addresses a **"fundamental challenge"** (YdhY) and tackles the **"critical and urgent problem"** of data contamination in LLM evaluation (WaAN). We are particularly pleased that our empirical results were highlighted as **"surprising and counter-intuitive"** (WaAN), challenging the common assumption that **"CoT equals better reasoning."** Reviewers noted that the identification of **"reverse scaling laws"** represents a **"significant discovery"** and a **"potentially highly impactful contribution"** (Jwi6) to the field.

**2. Novelty & Methodology:**
Our **"generalized puzzle-solving"** paradigm was praised as **"highly innovative"** (WaAN) and **"creative"** (ka3X), providing a **"more authentic assessment"** of knowledge-based reasoning (WaAN) that **"effectively mitigates data leakage"** (ka3X). Furthermore, specific components of our analysis, such as the incorrect guessing metric, were recognized as a **"real insight"** and a **"novel idea"** (Jwi6).

The reviewers' primary concerns focused on the validation of difficulty and experimental controls, which we have addressed with new topological analysis, human verification, and extensive ablation studies. The revisions are marked with blue text in the PDF. We summarize these key responses below:

**1. The Difficulty Definition and Detailed Data Construction Pipeline (YdhY, ka3X, Jwi6)**
We clarified our rigorous 0.14% coarse-to-fine filtering funnel, which distilled 6.4 million Wikipedia titles down to ~9,000 "tail" entities through four stages. To validate "doctoral-level" difficulty objectively, we conducted a topological analysis, finding that PEB targets have a median In-link count of 16 (specialized "edge nodes") compared to 500+ for common entities. This was further corroborated by manual verification from domain-expert PhD candidates.

**2. The Reliability of the Environment Agent Verification (WaAN, ka3X, Jwi6)**
We addressed concerns about evaluation stability by cross-validating the benchmark with a stronger Environment Agent (Claude Sonnet 4). Our analysis reveals a distinct "Rank Convergence" phenomenon: as sample size increases, inter-judge agreement strengthens significantly (Global Spearman $\rho$ improved from 0.86 to 0.96; Domain-level from 0.61 to 0.89). This confirms that the relative performance hierarchy（specifically the underperformance of "reasoning-enhanced" models）is a statistically robust signal rather than evaluation noise.

**3. Paradigm and Mechanical Dependency (WaAN, Jwi6)**
We validated that the "Reasoning Gap" is a structural failure robust to hyperparameter changes (e.g., DeepSeek R1 fails even at T=1.0). To clarify the practical significance of the paradigm, we adopted Reviewer Jwi6's suggestion to move the "Medical Diagnosis" parallel to the Introduction. This explicitly frames the benchmark's motivation as evaluating ontology-aware knowledge reasoning (similar to a doctor narrowing down a diagnosis) rather than mere game mechanics.

**4. Mathematical Analysis (YdhY, Jwi6)**
To enhance the theoretical rigor, we have explicitly formalized the core concept of "Ontological Traversal" in Section 3.1 of the revised paper. Furthermore, we added Appendix D as a theoretical supplement. This section mathematically models the puzzle-solving process as a search problem over a latent knowledge graph, using Information Gain formulation to provide a theoretical basis for the inefficient search trajectories observed in reasoning models.

---

### Meta-Review · Area_Chair_Wd5b · 2026-01-06

**Summary:**

The paper introduces the Puzzle Everything Benchmark (PEB), a framework evaluating reasoning over internalized knowledge by treating Wikipedia concepts as puzzles. While the motivation to address data leakage is well-founded, the reviewers were not convinced that the "puzzle-solving" paradigm effectively isolates reasoning capabilities from game-playing mechanics. The authors are commended for an extensive rebuttal. However, some concerns remain undecided in this round of submission.

**Reviewer Concerns:**

1. Concerns about the judge. The judge is still LLM based even in the new results during rebuttal

2. Paradigm lacks of generality as it is based on game-playing.

3. Some other subjective claims.

**Reviewer Scores:**

6/6/4/4

---

### Decision · Program_Chairs · 2026-01-26

Reject